# BDQL: Offline RL via Behavior Diffusion Q-learning without Policy Constraint

## Abstract

Offline reinforcement learning (RL) algorithms often constrain the policy or regularize the value function within an off-policy actor-critic framework to overcome the overestimation on out-of-distribution (OOD) actions. And the on-policy style offline algorithms also cannot escape from these constraints (or regularization). In this paper, we propose an on-policy style algorithm, **B**ehavior **D**iffusion **Q**-**L**earning (BDQL), which has the potential to solve offline RL without introducing any potential constraints. BDQL first recovers the behavior policy through the diffusion model and then updates this diffusion-based behavior policy using the behavior Q-function learned by SARSA. The update of BDQL exhibits a special two-stage pattern. At the beginning of the training, thanks to the precise modeling of the diffusion model, the on-policy guidance of the behavior Q-function over the behavior policy is effective enough to solve the offline RL. As training processes, BDQL suffers from the OOD issue, causing the training fluctuation or even collapse. Consequently, OOD issue arises after BDQL solves the offline problem which means the policy constraint is not necessary for solving offline RL in BDQL. Although the policy constraint can overcome the OOD issue and then completely address the training fluctuation, it also has a negative impact on solving the offline problem in the first stage. Therefore, we introduce the stochastic weight averaging (SWA) to mitigate the training fluctuation without affecting the offline solution. Experiments on D4RL demonstrate the special two-stage training phenomenon, where the first stage does have the capability to solve offline RL.

## 1 Introduction

Offline reinforcement learning (RL) (Levine et al., 2020) differs from traditional online reinforcement learning (Sutton et al., 1998) in that it only allows learning from a pre-collected offline dataset (Fu et al., 2020; Lange et al., 2012) and does not permit interaction with the environment. Off-policy learning in traditional RL learns from a replay buffer, which bears some resemblance to learning from the offline dataset. As a result, early researches attempt to apply the classical Actor-Critic (AC) framework (Sutton et al., 1999; Konda & Tsitsiklis, 1999; Degris et al., 2012) from off-policy learning to offline RL, but the performance is disappointing (Fujimoto et al., 2019). This is because in policy evaluation, the agents tend to poorly estimate the value of state-action pairs out of the offline dataset. The poor estimation in turn affects policy improvement, where agents choose out-of-distribution (OOD) actions with highly overestimated value. There are two main approaches to address this OOD issue: During the policy evaluation, regularizing the value function from assigning excessively high values to out-of-distribution state-action pairs (Kumar et al., 2020; Kostrikov et al., 2021; Bai et al., 2022). During the policy improvement, constraining the policy from deviating too far from the behavior policy (Fujimoto & Gu, 2021; Wu et al., 2019; Tarasov et al., 2023).

Some researches attempt to analyze offline RL from an on-policy learning perspective. These algorithms first learn a behavior policy using behavior cloning (BC) (Pomerleau, 1988) and then optimize this policy by the behavior Q-function learned via SARSA along with some constraints. R-BVE (Gulcehre et al., 2021) and Onestep RL (Brandfonbrener et al., 2021) transforms off-policy style algorithms into an on-policy form with all the regularization or constraints maintained. Starting from the offline monotonic policy improvement, BPPO (Zhuang et al., 2023) proposes that the classical online on-policy PPO (Schulman et al., 2017) is naturally able to solve offline RL since the "clip" operation of PPO is essentially the policy constraint through Total Variational (TV) Distance.

In this paper, we propose an on-policy style offline algorithm called **B**ehavior **D**iffusion **Q**-**L**earning (BDQL) which has the potential to solve offline RL **without** any policy constraints. First, we recover the behavior policy through behavior cloning (BC) (Pomerleau, 1988). The true behavior policy can encompass multiple policies, expert demonstrations or artificially designed planners (Fu et al., 2020), leading to complex multi-modal distribution in offline dataset (Wang et al., 2022; Shafiullah et al., 2022). This poses a significant challenge for precisely recovering the true behavior policy. Inspired by the success in continuous action modeling (Pearce et al., 2023; Wang et al., 2022), we adopt the diffusion (Ho et al., 2020) as the behavior policy. Then we calculate the behavior Q-function using SARSA (Rummery & Niranjan, 1994) and finally update the diffusion-based behavior policy by this fixed behavior Q-function using the deterministic policy gradient (DPG) (Silver et al., 2014).

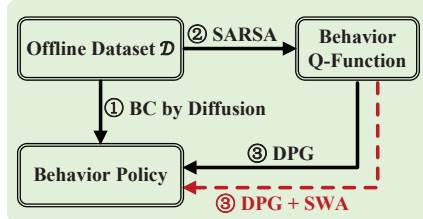

Figure 1: The overview of BDQL where DPG is the deterministic policy gradient and SWA is stochastic weight averaging.

*Why BDQL can solve offline RL **without** introducing policy **constraint** to address the OOD issue?*

To address this, we analyze if BDQL can converge to the optimal policy and its tenable condition. Specifically, this condition requires that the distance between the estimated behavior policy and the true one is sufficiently small, which highlights the role of diffusion in BC. Furthermore, we roughly divide the training into two stages based on the distance between the current training policy and the true behavior policy. The OOD issue only arises in the second stage when this distance becomes too large, leading to training fluctuations or even collapse. However, the early policy training adheres to the guarantee of optimality. This implies BDQL can solve offline RL before OOD occurs. Therefore, constraint is not mandatory for solving offline problem. Only the training fluctuation caused by OOD should be addressed. So we introduce stochastic weight averaging (SWA) Izmailov et al. (2018), which has made the online learning more stable (Nikishin et al., 2018). SWA won't change the actual optimization direction of BDQL, only stabilizing the fluctuation or mitigating the collapse.

Experiments on D4RL (Fu et al., 2020) indeed exhibits a two-phase phenomenon of the BDQL training process. The policy first ascends to high performance, followed by potential fluctuation or even collapse that results in a sharp performance drop. The BDQL peak performance of the first stage, namely the best result, is even comparable with ensemble-based methods (An et al., 2021). This indicates BDQL can solve the offline RL without the policy constraint before ODD occurs. However, training fluctuation or collapse leads to unsatisfactory last results. Constraint can completely overcome the fluctuation but the performance is unavoidably harmed since constraint conflicts with the behavior Q-function. In contrast, BDQL along with the SWA achieves a significant improvement in the last result, comparable to the classical baselines, with little best result decline.

## 2 PRELIMINARIES

### 2.1 OFFLINE REINFORCEMENT LEARNING

Reinforcement Learning (RL) is a framework of sequential decision. Typically, this problem is formulated by a Markov decision process (MDP) $\mathcal{M} = \{\mathcal{S}, \mathcal{A}, r, p, d_0, \gamma\}$, with state space $\mathcal{S}$, action space $\mathcal{A}$, scalar reward function $r$, transition dynamics $p$, initial state distribution $d_0(\mathbf{s}_0)$ and discount factor $\gamma$ (Sutton et al., 1998). The MDP is often called the environment and the agent is a policy, which predicts an action based on the current state $\mathbf{a}_t = \pi(\mathbf{s}_t)$, where $\mathbf{a}_t \in \mathcal{A}, \mathbf{s}_t \in \mathcal{S}$. When the policy interacts with the environment $\mathcal{M}$, a sequence full of states and actions called trajectory $\tau = (\mathbf{s}_0, \mathbf{a}_0, \cdots, \mathbf{s}_T, \mathbf{a}_T)$ is generated. The goal of RL is learn a policy that can maximize the expectation of the discounted accumulated return $J(\pi) = \mathbb{E}_\tau \left[ \sum_{t=0}^{T} \gamma^t r(\mathbf{s}_t, \mathbf{a}_t) \right]$. This objective can also be measured by a state-action value function, Q-function $Q(\mathbf{s}, \mathbf{a})$, the expected discounted accumulated return given the action $\mathbf{a}$ in state $\mathbf{s}$: $Q(\mathbf{s}, \mathbf{a}) = \mathbb{E}_\tau \left[ \sum_{t=0}^{T} \gamma^t r(\mathbf{s}_t, \mathbf{a}_t) | \mathbf{s}_0 = \mathbf{s}, \mathbf{a}_0 = \mathbf{a} \right]$. In **offline** RL (Levine et al., 2020), the agent only has access to a fixed dataset with transitions $\mathcal{D} = \left\{ (\mathbf{s}_t, \mathbf{a}_t, r_t, \mathbf{s}_{t+1}, \mathbf{a}_{t+1})_{t=1}^{T} \right\}$ collected by some arbitrary unknown process. For the sake of clarity and conciseness, we formally refer to this collect process as the behavior policy $\pi_b$. Offline RL expects the agent to infer a policy from the dataset without interacting with the environment $\mathcal{M}$.

## 2.2 ON-POLICY LEARNING AND OFF-POLICY LEARNING

In online reinforcement learning, based on the source of experience for policy updates, we can categorize the algorithm into two major classes: on-policy online RL and off-policy online RL. In on-policy learning, the policy first interacts with the environment to collect the experience and then uses the experience to improve itself. The following is a classic actor-critic (AC) framework in on-policy learning (Konda & Tsitsiklis, 1999; Sutton et al., 1999), which alternates between policy evaluation and policy improvement with the randomly initialized $Q_{\phi_0}, \pi_{\theta_0}$:

$$Q_{\phi_{k+1}} \leftarrow \arg \min_{\phi} \mathbb{E}_{(\mathbf{s}, \mathbf{a}, \mathbf{s}', \mathbf{a}') \sim \pi_{\theta_k}} \left[ \left( r\left(\mathbf{s}, \mathbf{a}\right) + \gamma Q_{\phi_k}\left(\mathbf{s}', \mathbf{a}'\right) - Q_{\phi}\left(\mathbf{s}, \mathbf{a}\right) \right)^2 \right],$$

$$\pi_{\theta_{k+1}} \leftarrow \arg \max_{\theta} \mathbb{E}_{\mathbf{s} \sim \pi_{\theta_k}, \mathbf{a} \sim \pi_{\theta}} \left[ Q_{\phi_{k+1}}\left(\mathbf{s}, \mathbf{a}\right) \right],$$

here both $(\mathbf{s}, \mathbf{a}, \mathbf{s}', \mathbf{a}') \sim \pi_{\theta_k}$ and $\mathbf{s} \sim \pi_{\theta_k}$ denote the experience is only from the policy $\pi_{\theta_k}$. While for off-policy learning, the experience used to improve the policy originates not only from itself $\pi_{\theta_k}$ but can also other polices $\pi_{\theta_{k-1}}, \pi_{\theta_{k-2}}, \cdots, \pi_{\theta_0}$. All the experience composes the online replay buffer $\mathcal{D}^+$ (Degris et al., 2012). Then the AC framework evolves into the following off-policy style:

$$Q_{\phi_{k+1}} \leftarrow \arg \min_{\phi} \mathbb{E}_{(\mathbf{s}, \mathbf{a}, \mathbf{s}') \sim \mathcal{D}_+, \mathbf{a}' \sim \pi_{\theta_k}} \left[ \left( r\left(\mathbf{s}, \mathbf{a}\right) + \gamma Q_{\phi_k}\left(\mathbf{s}', \mathbf{a}'\right) - Q_{\phi}\left(\mathbf{s}, \mathbf{a}\right) \right)^2 \right], \quad (1)$$

$$\pi_{\theta_{k+1}} \leftarrow \arg \max_{\theta} \mathbb{E}_{\mathbf{s} \sim \mathcal{D}_+, \mathbf{a} \sim \pi_{\theta}} \left[ Q_{\phi_{k+1}}\left(\mathbf{s}, \mathbf{a}\right) \right]. \quad (2)$$

While both offline RL and online off-policy learning learn from a pre-collected collection of experiences ($\mathcal{D}$ or $\mathcal{D}_+$), naively applying the off-policy actor-critic framework to offline RL can lead to poor performance (Fujimoto et al., 2019; Levine et al., 2020). Specifically, Equation 1 may query the estimated $Q_{\phi_k}\left(\mathbf{s}', \mathbf{a}'\right)$ on out-of-distribution (OOD) actions that lie far away from the offline dataset $\mathcal{D}$, resulting in pathological value $Q_{\phi_{k+1}}\left(\mathbf{s}, \mathbf{a}\right)$ that incurs large error. Furthermore, this error will cause the inferred policy $\pi_{\theta_{k+1}}$ to be biased towards OOD actions with erroneously overestimated values but actually with low rewards. For simplicity, we summarize the problem of offline RL as the overestimation of OOD actions and even abbreviates it to "OOD issue" when understanding is clear.

## 2.3 DIFFUSION MODEL

Since both reinforcement learning and diffusion process share the concept of timesteps, we use subscripts $t \in \{1, \ldots, T\}$ to denote trajectory timestep in RL and superscripts $n \in \{1, \ldots, N\}$ to denote diffusion timestep. Diffusion-based models (Ho et al., 2020; Sohl-Dickstein et al., 2015; Song & Ermon, 2019) are a special kind of generative models which learns the data distribution $q(x)$ from a distribution $p\left(\boldsymbol{x}^0\right)$. Diffusion models assume $p_\theta\left(\boldsymbol{x}^0\right) := \int p_\theta\left(\boldsymbol{x}^{0:N}\right) d\boldsymbol{x}^{1:N}$, where $\boldsymbol{x}^1, \ldots, \boldsymbol{x}^N$ are latent variables of the same dimensionality as the data $\boldsymbol{x}^0 \sim p\left(\boldsymbol{x}^0\right)$. A forward diffusion chain gradually adds noise to the data $\boldsymbol{x}^0 \sim q\left(\boldsymbol{x}^0\right)$ in $N$ steps with a pre-defined variance schedule $\beta^n$, expressed as $q\left(\boldsymbol{x}^{1:N}|\boldsymbol{x}^0\right) := \prod_{n=1}^N q\left(\boldsymbol{x}^n|\boldsymbol{x}^{n-1}\right) = \prod_{n=1}^N \mathcal{N}\left(\boldsymbol{x}^n; \sqrt{1-\beta^n}\boldsymbol{x}^{n-1}, \beta^n \boldsymbol{I}\right)$. On the contrary, the reverse diffusion chain denoises and is constructed as $p_\theta\left(\boldsymbol{x}^{0:N}\right) := \mathcal{N}\left(\boldsymbol{x}^N; \boldsymbol{0}, \boldsymbol{I}\right) \prod_{n=1}^N p_\theta\left(\boldsymbol{x}^{n-1}|\boldsymbol{x}^n\right)$. Starting with the Gaussian noise, samples are then iteratively generated through the reverse denoising process. The diffusion model is then optimized by maximizing the evidence lower bound defined as $\mathbb{E}_q\left[\ln p_\theta\left(\boldsymbol{x}^{0:N}\right) - \ln q\left(\boldsymbol{x}^{1:N}|\boldsymbol{x}^0\right)\right]$ (Blei et al., 2017; Jordan et al., 1999). After training, sampling from the diffusion model consists of sampling $\boldsymbol{x}^N \sim p\left(\boldsymbol{x}^N\right)$ and running the reverse diffusion chain to go from $n = N$ to $n = 0$. Diffusion models can be straightforwardly extended to conditional models by conditioning $p_\theta\left(\boldsymbol{x}^{n-1}|\boldsymbol{x}^n, c\right)$.

## 3 BEHAVIOR DIFFUSION Q-LEARNING

Offline RL has traditionally been studied from an off-policy perspective. Various policy constraints or value regularization are introduced to the off-policy AC to address the OOD issue (Kumar et al., 2020; Fujimoto & Gu, 2021). Some works have attempted to study offline RL from an on-policy perspective (Brandfonbrener et al., 2021; Gulcehre et al., 2021; Zhuang et al., 2023) and these on-policy style methods still cooperate with constraints or regularization although the constraint may inherently originate from the online method (Zhuang et al., 2023). In this paper, we propose an on-policy style algorithm, which has the potential to solve offline RL without introducing constraints.

**Behavior Diffusion Policy**    Similar to previous on-policy style methods, the first step is to recover an estimated behavior policy $\pi_\theta$ from the offline dataset $\mathcal{D}$. But the distribution of offline dataset often exhibit highly complex characteristics, such as skewness and multi-modality (Wang et al., 2022; Shafiullah et al., 2022), which may pose an obstacle in accurately modeling the behavior policy. Inspired by the tremendous success in modeling the continuous action spaces (Wang et al., 2022; Hansen-Estruch et al., 2023) and human behavior (Pearce et al., 2023), we introduce the diffusion model (Ho et al., 2020) to estimate the behavior policy $\mathbf{a}_t = \pi_\theta(\mathbf{s}_t)$. Concretely, the action $\mathbf{a}_t$ is modeled via the reverse process $p_\theta$ conditioned on the corresponding state $\mathbf{s}_t$:

$$p_\theta\left(\mathbf{a}_t^{0:N}|\mathbf{s}_t\right) = \mathcal{N}\left(\mathbf{a}_t^N; \mathbf{0}, \boldsymbol{I}\right) \prod_{n=1}^{N} p_\theta\left(\mathbf{a}_t^{n-1}|\mathbf{a}_t^n, \mathbf{s}_t\right). \tag{3}$$

The end sample of the reverse chain, $\mathbf{a}_t^0$, is the action used in RL while others are noised actions. We follow the DDPM (Ho et al., 2020) to represent the reverse process $p_\theta\left(\mathbf{a}_t^{n-1}|\mathbf{a}_t^n, \mathbf{s}_t\right)$ as a Gaussian distribution with a conditional noise prediction model $\epsilon_\theta$:

$$p_\theta\left(\mathbf{a}_t^{n-1}|\mathbf{a}_t^n, \mathbf{s}_t\right) = \mathcal{N}\left(\mathbf{a}_t^{n-1}; \boldsymbol{\mu}_\theta\left(\mathbf{a}_t^n, \mathbf{s}_t, n\right), \boldsymbol{\Sigma}_\theta\left(\mathbf{a}_t^n, \mathbf{s}_t, n\right)\right) \tag{4}$$

$$= \mathcal{N}\left(\mathbf{a}_t^{n-1}; \frac{1}{\sqrt{\alpha^n}}\left(\mathbf{a}_t^n - \frac{\beta^n}{\sqrt{1-\bar{\alpha}^n}}\epsilon_\theta\left(\mathbf{a}_t^n, \mathbf{s}_t, n\right)\right), \beta^n \boldsymbol{I}\right).$$

We first sample $\mathbf{a}_t^N \sim \mathcal{N}(\mathbf{0}, \boldsymbol{I})$ and then denote the reverse diffusion chain parameterized by $\theta$ as

$$\mathbf{a}_t^{n-1}|\mathbf{a}_t^n = \frac{\mathbf{a}_t^n}{\sqrt{\alpha^n}} - \frac{\beta^n}{\sqrt{\alpha^n\left(1-\bar{\alpha}^n\right)}}\epsilon_\theta\left(\mathbf{a}_t^n, \mathbf{s}_t, n\right) + \sqrt{\beta^n}\epsilon, \epsilon \sim \mathcal{N}(\mathbf{0}, \boldsymbol{I}), \text{ for } n = N, \ldots, 1. \tag{5}$$

Following DDPM (Ho et al., 2020), when $n = 1, \epsilon$ is set as $\mathbf{0}$ to improve the sampling quality. We minimize the simplified objective of DDPM to recover the behavior policy $\pi_\theta$:

$$\pi_\theta = \arg\min_\theta \mathbb{E}_{n \sim \mathcal{U}, \epsilon \sim \mathcal{N}(\mathbf{0},\boldsymbol{I}), (\mathbf{s}_t, \mathbf{a}_t) \sim \mathcal{D}}\left[\left\|\epsilon - \epsilon_\theta\left(\sqrt{\bar{\alpha}^n}\mathbf{a}_t + \sqrt{1-\bar{\alpha}^n}\epsilon, \mathbf{s}_t, n\right)\right\|^2\right], \tag{6}$$

where $\mathcal{U}$ is a uniform distribution over the discrete set as $\{1, \ldots, N\}$.

The diffusion-based policy can be efficiently optimized by sampling a single diffusion step $n$ for each $(\mathbf{s}_t, \mathbf{a}_t)$ pair since the calculation of forward process is non-iterative. But the reverse sampling which requires iteratively computing $\epsilon_\theta$ networks $N$ times, which is the main bottleneck for the running time. In order to improve efficiency, also reduce the value of $N$, we adopt the following format of $\beta^n = 1 - \alpha^n = 1 - e^{-\beta^{\min}(1/N) - 0.5\left(\beta^{\max} - \beta^{\min}\right)(2n-1)/N^2}$, where $\beta^{\min} = 0.1$ and $\beta^{\max} = 10.0$. This noise schedule is derived under the variance preserving SDE (Song et al., 2020).

**Behavior Q-function**    The experience within the offline dataset $\mathcal{D}$ can be viewed as the on-policy samples of the behavior policy. So we regard the Q-function directly calculated via SARSA (Rummery & Niranjan, 1994) as an behavior Q-function:

$$Q_\phi = \arg\min_\phi \mathbb{E}_{(\mathbf{s}_t, \mathbf{a}_t, r_t, \mathbf{s}_{t+1}, \mathbf{a}_{t+1}) \sim \mathcal{D}}\left[\left(r_t + \gamma\bar{Q}_\phi\left(\mathbf{s}_{t+1}, \mathbf{a}_{t+1}\right) - Q_\phi\left(\mathbf{s}_t, \mathbf{a}_t\right)\right)^2\right], \tag{7}$$

where $\bar{Q}_\phi$ is target network of $Q_\phi$.

**Fixed Q-function Update**    Finally, we update the estimated diffusion behavior policy $\pi_\theta$ by maximizing the fixed behavior Q-function $Q_\phi$. The output of the diffusion policy is the deterministic action rather than the distribution of action, so the update is based on the deterministic policy gradient (DPG) theorem (Silver et al., 2014):

$$\pi_{\theta*} = \arg\max_\theta \mathbb{E}_{\mathbf{s}_t \sim \mathcal{D}}\left[Q_\phi\left(\mathbf{s}_t, \pi_\theta\left(\mathbf{s}_t\right)\right)\right]. \tag{8}$$

Here the Q-function $Q_\phi$ corresponds to the the behavior policy, so strictly speaking, the policy that takes the action should be $\pi_b$ rather than the estimated behavior policy $\pi_\theta$ from Equation (6). In the following section, we will analyze the error introduced by this approximation (also the difference between $\pi_b$ and $\pi_\theta$) and then highlight the role of precise behavior modeling by diffusion model. Our method is an on-policy style algorithm that updates the diffusion-based behavior policy using the fixed behavior Q-function, so we name it as **Behavior Diffusion Q-Learning (BDQL)**.

## 4 ANALYSIS

In this section, we analyze our algorithm from the theoretical perspective. For the sake of convenience in the proof, we rewrite Equation (8) in the following form:

$$\pi_{\theta^*} = \arg\max_{\pi_\theta} \widehat{J}(\pi_\theta) = \arg\max_{\pi_\theta} \mathbb{E}_{\mathbf{s}_t \sim \mathcal{D}, \mathbf{a}_t \sim \pi_\theta} [Q_{\mathrm{b}}(\mathbf{s}_t, \mathbf{a}_t)], \tag{9}$$

here the fixed behavior Q-function $Q_{\mathrm{b}}$ updates the estimated behavior policy $\pi_\theta$. An ideal and rigorous formulation should demand the action is taken by the true behavior policy $\pi_{\mathrm{b}}$:

$$J(\pi_{\mathrm{b}}) = \mathbb{E}_{\mathbf{s}_t \sim \mathcal{D}, \mathbf{a}_t \sim \pi_{\mathrm{b}}} [Q_{\mathrm{b}}(\mathbf{s}_t, \mathbf{a}_t)], \tag{10}$$

where $Q_{\mathrm{b}}$ is also fixed. Then we will demonstrate the optimality of the theoretical form $J(\pi_{\mathrm{b}})$ and analyze the gap between $J(\pi_{\mathrm{b}})$ and the BDQL practical form $\widehat{J}(\pi_{\mathrm{b}})$. Concretely, the theoretical formulation eventually derives an policy whose value function can recover the optimal one in offline RL. And the gap between $J(\pi_{\mathrm{b}})$ and $\widehat{J}(\pi_{\mathrm{b}})$ relates to the distance between policies $\pi_{\mathrm{b}}$ and $\pi_\theta$.

**Optimality of $J(\pi_{\mathrm{b}})$:** We denote the policies obtained by maximizing $J(\pi_{\mathrm{b}})$ as $\pi_{\mathrm{b}}, \pi_{\mathrm{b}'}, \pi_{\mathrm{b}''}, \cdots, \pi_{\mathrm{b}^*}$ respectively. To demonstrate the optimality of $\pi_{\mathrm{b}^*}$, we introduce the support-constraint value function (Kostrikov et al., 2021; Kumar et al., 2019; Wu et al., 2022), where the action distribution is constrained to the behavior policy: $\mathbb{V}_\pi(\mathbf{s}) = \mathbb{E}_{\substack{\mathbf{a} \sim \pi \\ \text{s.t. } \pi_{\mathrm{b}}(\mathbf{a}|\mathbf{s}) > 0}} Q_\pi(\mathbf{s}, \mathbf{a})$.

**Theorem 1.** *For any state $\mathbf{s}$, the support-constraint value function of $\pi_{\mathrm{b}^*}$ converges to the offline optimal Q-function:*

$$\mathbb{V}_{\pi_{\mathrm{b}^*}}(\mathbf{s}) \longrightarrow \max_{\substack{\mathbf{a} \in \mathcal{A} \\ \text{s.t. } \pi_{\mathrm{b}}(\mathbf{a}|\mathbf{s}) > 0}} Q^*(\mathbf{s}, \mathbf{a}), \tag{11}$$

*where $Q^*(\mathbf{s}, \mathbf{a})$ is an optimal support-constraint Q-function and defined as*

$$Q^*(\mathbf{s}, \mathbf{a}) = r(\mathbf{s}, \mathbf{a}) + \gamma \mathbb{E}_{\mathbf{s}' \sim p(\cdot|\mathbf{s}, \mathbf{a})} \left[ \max_{\substack{\mathbf{a}' \in \mathcal{A} \\ \text{s.t. } \pi_{\mathrm{b}}(\mathbf{a}'|\mathbf{s}') > 0}} Q^*(\mathbf{a}', \mathbf{s}') \right] \tag{12}$$

*Proof Sketch.* We first show that the support-constraint value functions of all policies do not surpass the optimal support-constraint Q-function $\mathbb{V}_{\pi_{\mathrm{b}}}(\mathbf{s}), \mathbb{V}_{\pi_{\mathrm{b}'}}(\mathbf{s}), \mathbb{V}_{\pi_{\mathrm{b}''}}(\mathbf{s}), \cdots, \mathbb{V}_{\pi_{\mathrm{b}^*}}(\mathbf{s}) \le \max_{\substack{\mathbf{a} \in \mathcal{A} \\ \text{s.t. } \pi_{\mathrm{b}}(\mathbf{a}|\mathbf{s}) > 0}} Q^*(\mathbf{s}, \mathbf{a})$. Then, we extend policy improvement from online to offline scenarios to prove the improvement of the support-constraint value function $\mathbb{V}_{\pi_{\mathrm{b}}}(\mathbf{s}) \le \mathbb{V}_{\pi_{\mathrm{b}'}}(\mathbf{s}), \mathbb{V}_{\pi_{\mathrm{b}'}}(\mathbf{s}) \le \mathbb{V}_{\pi_{\mathrm{b}''}}(\mathbf{s}), \cdots$. Combining these two points completes the proof. See Appendix A for details. □

**Gap between $J(\pi_{\mathrm{b}})$ and $\widehat{J}(\pi_{\mathrm{b}})$:** In the above Theorem 1, we have demonstrated the optimality of policy obtained by maximizing the theoretical form $J(\pi_{\mathrm{b}})$. Now we analyze the gap $\left| J(\pi_{\mathrm{b}}) - \widehat{J}(\pi_\theta) \right|$ and find this gap relates to the distance between $\pi_{\mathrm{b}}$ and $\pi_\theta$. In the following theorem, we use the Total Varitional distance $D_{TV}(\pi_\theta \| \pi_{\mathrm{b}})[\mathbf{s}] = \frac{1}{2}\mathbb{E}_{\mathbf{a}} |\pi_\theta(\mathbf{a}|\mathbf{s}) - \pi_{\mathrm{b}}(\mathbf{a}|\mathbf{s})|$ for analysis.

**Theorem 2.** *The Gap between the theoretical form (10) the practical implementation in (9) can be bounded by the Total Varitional distance between the true and estimated behavior policies:*

$$\left| J(\pi_{\mathrm{b}}) - \widehat{J}(\pi_\theta) \right| \le 2 \cdot \mathcal{C}_{\mathcal{D}}^{\pi_{\mathrm{b}}} \cdot \boxed{\mathbb{E}_{\mathbf{s}_t \sim \mathcal{D}} \left[ D_{TV}(\pi_\theta \| \pi_{\mathrm{b}})[\mathbf{s}_t] \right]}, \tag{13}$$

here $\mathcal{C}_{\mathcal{D}}^{\pi_{\mathrm{b}}} = \max_{\mathbf{s}_t \sim \mathcal{D}, \mathbf{a}_t \in \mathcal{A}} Q_{\mathrm{b}}(\mathbf{s}_t, \mathbf{a}_t)$ is a constant and the proof is presented in Appendix B.

Combining above Theorem 1 and 2, we can conclude that:

> **Conclusion**
>
> The support-constraint value of policy $\pi_{\theta^*}$ derived from the BDQL update (8) will approach the optimal offline value function ($\mathbb{V}_{\pi_{\theta^*}}(\mathbf{s}) \longrightarrow \max_{\substack{\mathbf{a} \in \mathcal{A} \\ \text{s.t. } \pi_{\mathrm{b}}(\mathbf{a}|\mathbf{s}) > 0}} Q^*(\mathbf{s}, \mathbf{a})$) when and only when the distance is small enough ($\mathbb{E}_{\mathbf{s}_t \sim \mathcal{D}} \left[ D_{TV}(\pi_\theta \| \pi_{\mathrm{b}})[\mathbf{s}_t] \right] \longrightarrow 0$).

Based on this conclusion, the role of the diffusion model in BDQL can be more clearly highlighted. Compared to traditional behavior cloning, the diffusion policy can more accurately model the offline dataset, that is, a smaller distance $D_{TV}\left(\pi_\theta \| \pi_\mathrm{b}\right)$, which furthermore ensures the optimality. This is the unique role of the diffusion behavior policy under the BDQL framework.

**Two training stages of BDQL**  We denote the intermediate policy during BDQL training as $\pi_{\tilde{\theta}}$. Then the training process can be roughly divided into two stages according to the distance $D_{TV}\left(\pi_{\tilde{\theta}} \| \pi_\mathrm{b}\right)$. During the early training stage, the distance $D_{TV}\left(\pi_{\tilde{\theta}} \| \pi_\mathrm{b}\right)$ remains at a relatively small level, ensuring the guarantee of optimality as **Conclusion** 4 claimed. As training processes, the policy gradually deviates from the initial policy $\pi_\theta$ and the distance $D_{TV}\left(\pi_{\tilde{\theta}} \| \pi_\mathrm{b}\right)$ starts to increase, which results in a gradual weakening of the guarantee of optimality. Simultaneously, the policy indeed generates the OOD state-action pairs, and the fixed Q-function may not provide accurate estimation of them. This error won't be accumulated and amplified since the Q-function is no longer updated. But the OOD issue may cause the training fluctuation or even collapse at last.

We can employ any policy constraint to address the OOD issue, thus completely resolving the potential training fluctuation. For example, we can add additional behavior cloning term in Equation (8), which is similar to the Diffusion Q-Learning (Wang et al., 2022) or TD3+BC (Fujimoto & Gu, 2021). However, such constraint would inevitably pose a negative impact on the optimality of the first training stage. It seems like we are caught in a dilemma where we cannot address the OOD issue while retaining the optimality. One question thus arises:

*Does the OOD problem really need to be addressed during the learning process of BDQL?*

**Not necessarily!** BDQL's early training stage does not encounter OOD issues and exhibits strong optimality guarantees. Thus, BDQL has the potential to obtain a policy capable of addressing offline RL **before** the occurrence of OOD, which is the unique characteristic of BDQL. Therefore, we only need to address the potential training fluctuations caused by OOD. Stochastic Weight Averaging (SWA) can only stabilize the training from fluctuation and not alter the early training.

Stochastic Weight Averaging (SWA) averages the multiple checkpoints during the optimization. After training for a certain number of steps $K$, SWA equally averages the checkpoints every $c$ steps:

$$\pi_{\theta_k}^{\mathrm{SWA}} = \frac{\pi_{\theta_{k-c}}^{\mathrm{SWA}} \cdot \frac{k-K}{c} + \pi_{\theta_k}}{\frac{k-K}{c}+1}, \text{if} \mod (k-K,c)=0, \tag{14}$$

where the initial SWA checkpoint is $\pi_{\theta_k}^{\mathrm{SWA}} = \pi_{\theta_K}$. At the next training step $k+1$, the Equation 8 still updates the original checkpoint $\pi_{\theta_k}$ rather than the SWA checkpoint $\pi_{\theta_k}^{\mathrm{SWA}}$. In other words, SWA checkpoints do not participate in training. They are only the average of the original checkpoints throughout the whole training process. This implies that SWA does not alter the optimization direction of the original BDQL, which is different from any constraint.

However, SWA can significantly alleviate the training fluctuation. If the checkpoint $\pi_{\theta_k}$ experiences intense training fluctuation, the performance of the $\pi_{\theta_k}^{\mathrm{SWA}}$ won't plummet dramatically thanks to the presence of the previously checkpoints. As subsequent checkpoints becomes stable, the performance fluctuation of the the checkpoint $\pi_{\theta_k}$ becomes negligible in some degree. This demonstrates that SWA can significantly stabilize the entire training process. Sometimes, the performance of BDQL first rises and then continues to decline, that is, BDQL suffers from the training collapse. The presence of previous high-performance checkpoints can relatively averages the latter performance declined checkpoints, mitigating the onset of the performance collapse phase.

## 5    RELATED WORK

**Offline Reinforcement Learning**  Classical online off-policy actor-critic framework (Konda & Tsitsiklis, 1999; Sutton et al., 1999; Degris et al., 2012) often fails in offline RL due to the distribution shift (Levine et al., 2020) or extrapolation error (Fujimoto et al., 2019). Offline RL methods within this framework be divided into two categories. One category is policy constraint, which constrains the learned policy stay close to the behavior policy. A variety of methods are proposed based on different "distance" such as batch constrained (Fujimoto et al., 2019), KL divergence (Wu et al., 2019), MMD distance (Kumar et al., 2019) and MSE constraint (Fujimoto & Gu, 2021). Another

category is value regularization, which regularizes the value function to assign low values on OOD state-action pairs (Kumar et al., 2020; Kostrikov et al., 2021; Bai et al., 2022). The strongest set of baselines is the ensemble-based methods (An et al., 2021; Yang et al., 2022; Ghasemipour et al., 2022), where the Q-function is estimated by the minimum of a large number of Q-networks.

Some offline methods understands and solve offline RL from the perspective of on-policy learning. R-BVE (Gulcehre et al., 2021) and Onestep RL (Brandfonbrener et al., 2021) both transform off-policy algorithms (such as CRR (Wang et al., 2020), BCQ (Fujimoto et al., 2019), BRAC (Wu et al., 2019)) into on-policy style with constraint or regularization remained, where the estimated behavior policy is updated by the behavior Q-function learned through SARSA (Rummery & Niranjan, 1994). BPPO (Zhuang et al., 2023) finds that online on-policy method PPO (Schulman et al., 2017) can naturally solve offline RL and the only modification is the calculation of advantage function. This is because the clip operation of PPO has a strong connection with policy constraint with Total Varitional Distance. The success of these on-policy style methods also depends on the constraint. We propose BDQL, an on-policy style offline RL algorithm but no policy constraint is introduced.

**Diffusion Model in Reinforcement Learning** Diffusion model (Ho et al., 2020) has been introduced into offline RL to model different distributions, including the trajectories, states or actions. Diffuser (Janner et al., 2022) models the trajectory distribution with unconditional diffusion model and requires a trained reward function on noisy state-action pairs. Decision Diffuser (Ajay et al., 2022) only models the distribution of the states sequence by diffusion model conditioned on the return, skills or constraints and trains another inverse dynamics model to recover the actions from states sequence. In the last category, diffusion model serves as the policy. Diffusion policy can enhance the offline algorithms such as Diffusion Q-Learning (Wang et al., 2022) and IDQL (Hansen-Estruch et al., 2023), accurately model the human behavior (Pearce et al., 2023) or serve as goal-conditioned policy for imitation learning (Reuss et al., 2023). In Select from Behavior Candidates (SfBC) (Chen et al., 2022), the policy is defined as diffusion model with importance reweighting. The behavior policy in BDQL is implemented by the conditional diffusion model and trained by BC.

## 6 EXPERIMENTS

We conduct a series of experiments on D4RL Gym (v2) tasks (Fu et al., 2020) to 1) evaluate the theoretical and practical performance of BDQL compared with classical, advanced and ensemble-based baselines, 2) analyze the role of each design components including diffusion policy, SWA and *No* policy constraint 3) and demonstrate the special two-stage training phenomenon of BDQL where the first stage is indeed able to solve the offline RL with no policy constraint introduced.

Table 1: The normalized best and last results on D4RL Gym tasks. We **bold** the best results and the results of D-BC, BDQL, BDQL-SWA are calculated by averaging the mean returns over 10 evaluation trajectories and five random seeds. (Abbreviations: HalfCheetah → HC, Hopper → HP, Walker2d → WK, medium → m, medium-replay → mr, medium-expert → me.)

| Tasks | Best Result | | | | | | | | | | | |
|---|---|---|---|---|---|---|---|---|---|---|---|---|
| | BC | TD3+BC | AWAC | CQL | IQL | D-QL | ReBRAC | SAC-N | EDAC | D-BC | BDQL | BDQL-SWA |
| HC-m | 43.6 | 48.9 | 50.06 | 47.62 | 48.84 | 51.50 | 65.62 | **72.21** | 69.72 | 42.11±0.39 | 53.09±0.28 | 52.37±0.15 |
| HC-mr | 40.52 | 45.84 | 46.35 | 46.43 | 45.35 | 48.30 | 52.22 | **67.29** | 66.55 | 35.85±2.45 | 44.74±0.60 | 44.09±0.17 |
| HC-me | 79.69 | 96.59 | 96.11 | 97.04 | 95.38 | 97.20 | 108.89 | **111.73** | 110.62 | 48.84±6.93 | 96.36±0.16 | 95.59±0.28 |
| HP-m | 69.04 | 70.44 | 97.9 | 70.8 | 80.46 | 96.60 | **103.19** | 101.79 | 103.26 | 48.42±3.51 | **103.54±0.43** | 102.84±0.33 |
| HP-mr | 68.88 | 98.12 | 100.91 | 101.63 | 102.69 | 102.00 | 102.57 | **103.83** | 103.28 | 24.93±8.39 | **103.78±0.37** | 102.93±1.31 |
| HP-me | 90.63 | **113.22** | 103.82 | 112.84 | 113.18 | 112.30 | 113.16 | 111.24 | 111.8 | 56.36±7.69 | 112.85±0.40 | 111.23±1.71 |
| WK-m | 80.64 | 86.91 | 83.37 | 84.77 | 87.58 | 87.30 | 87.79 | 90.17 | **95.78** | 66.48±2.41 | 91.58±0.45 | 90.10±1.42 |
| WK-mr | 48.41 | 91.17 | 86.51 | 89.39 | 89.94 | 98.00 | 91.11 | 85.18 | 89.69 | 24.46±8.96 | **103.94±1.58** | 101.13±1.76 |
| WK-me | 109.95 | 112.21 | 108.28 | 111.63 | 113.06 | 111.20 | 112.49 | **116.93** | 116.52 | 94.32±6.68 | 116.51±1.07 | 116.56±1.39 |
| *Average* | *70.15* | *84.83* | *85.92* | *84.68* | *86.28* | *89.30* | *93.00* | *95.00* | *96.36* | *49.05* | *91.82* | *90.76* |

| Tasks | Last Result | | | | | | | | | | | |
|---|---|---|---|---|---|---|---|---|---|---|---|---|
| | BC | TD3+BC | AWAC | CQL | IQL | D-QL | ReBRAC | SAC-N | EDAC | D-BC | BDQL | BDQL-SWA |
| HC-m | 42.4 | 48.1 | 49.46 | 47.04 | 48.31 | 51.10 | 64.04 | **68.2** | 67.7 | 42.11±0.39 | 52.23±0.38 | 51.87±0.14 |
| HC-mr | 35.66 | 44.84 | 44.7 | 45.04 | 44.46 | 47.80 | 51.18 | 60.7 | **62.06** | 35.85±2.45 | 42.95±0.98 | 42.74±0.32 |
| HC-me | 55.95 | 90.78 | 93.62 | 95.63 | 94.74 | 96.80 | 103.8 | 98.96 | **104.76** | 48.84±6.93 | 93.79±1.03 | 93.73±1.51 |
| HP-m | 53.51 | 60.37 | 74.45 | 59.08 | 67.53 | 90.50 | **102.29** | 40.82 | 101.7 | 48.42±3.51 | 86.00±9.51 | **102.30±0.54** |
| HP-mr | 29.81 | 64.42 | 96.39 | 95.11 | 97.43 | **101.30** | 94.98 | 100.33 | 99.66 | 24.93±8.39 | 99.31±4.46 | 96.80±4.88 |
| HP-me | 52.3 | 101.17 | 52.73 | 99.26 | 107.42 | **111.10** | 109.45 | 101.31 | 105.19 | 56.36±7.69 | 37.28±8.01 | 71.36±16.72 |
| WK-m | 63.23 | 82.71 | 66.53 | 80.75 | 80.91 | 87.00 | 85.82 | 87.47 | 93.36 | 66.48±2.41 | 80.70±3.72 | 86.72±1.91 |
| WK-mr | 21.8 | 85.62 | 82.2 | 73.09 | 82.15 | **95.50** | 84.25 | 78.99 | 87.1 | 24.46±8.96 | 73.25±21.57 | 85.06±5.59 |
| WK-me | 98.96 | 110.03 | 49.41 | 109.56 | 111.72 | 110.10 | 111.86 | **114.93** | 114.75 | 94.32±6.68 | 113.88±1.42 | **115.00±1.65** |
| *Average* | *50.4* | *76.45* | *67.72* | *78.28* | *81.63* | *88.00* | *89.74* | *83.52* | *92.92* | *49.05* | *75.48* | *82.8* |

## 6.1 BENCHMARK RESULTS

To comprehensively evaluate the performance of BDQL, we assess it from both the best and last results. The best results are obtained through online evaluation while the last results are the last training checkpoint. We compare BDQL with a wide range of baselines, including classical, advanced, and ensemble-based baselines. Classical baselines are BC (Pomerleau, 1988), TD3+BC (Fujimoto & Gu, 2021), AWAC (Nair et al., 2020), CQL (Kumar et al., 2020), and IQL (Kostrikov et al., 2021). These algorithms are proposed relatively early and each one represents a classic idea. Advanced baselines are Diffusion Q-Learning (D-QL) (Wang et al., 2022) and ReBRAC (Tarasov et al., 2023). The advanced baselines offer more sophisticated tricks compared to the basic ones. D-QL replaces the behavior cloning term in TD3+BC with the diffusion model and ReBRAC incorporates numerous effective implementations on top of BRAC (Wu et al., 2019). Ensemble-based baselines are SAC-N and EDAC (An et al., 2021), which are the strongest class of offline baselines. These methods approximate the true Q value using the minimum from a large number of Q networks. The D-QL results are extracted from its appendix while others are from the Clean Offline Reinforcement Learning (CORL) that aims to provide fair comparison (Tarasov et al., 2022).

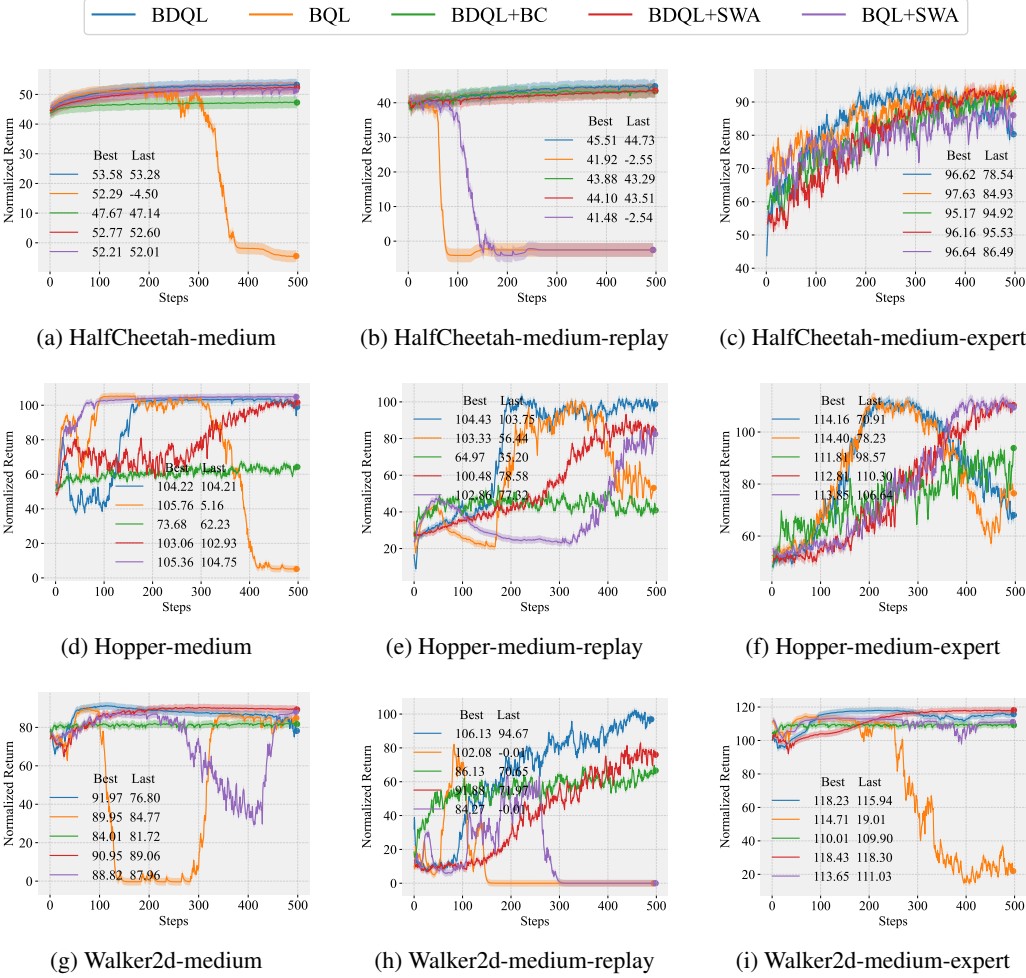

Figure 2: The learning curves of BDQL, BQL, BDQL+BC, BDQL+SWA and BQL+SWA. In each sub-figures, we list the best and last results of these by turn. The total training steps is 50,000 with an evaluation performed every 100 steps. Therefore, the step on the x-axis of the figure is 500.

The column D-BC in Table 1 is the last checkpoint of the diffusion-based behavior policy of BDQL, so we list it on the left side of BDQL. ***Training Fluctuation:*** Calculating the percentage decrease from best to last results, we find that BDQL experiences 17.8% decrease, only better than BC and AWAC. This substantial decrease indicates that BDQL indeed suffers from training fluctuation or

even collapse. ***Best Results:*** However, BDQL demonstrates exceptional best results. In environment `HalfCheetah`, BDQL is better than the classical baselines and D-QL. In environment `Hopper` and `Walker2d`, BDQL has achieved the best performance and even surpasses all the baselines on `Walker2d-medium-replay` tasks. ***Last Results:*** With the help of SWA, the fluctuation caused by OOD issue has been significantly alleviated. And the last results of BDQL-SWA is better than all the classical baselines with OOD issue remained, which is quite amazing and interesting.

## 6.2 ABLATION STUDY

We demonstrate the learning curves of BDQL and its four variants on Gym tasks in Figure 2. BDQL is the basic setting where the diffusion-based behavior policy is updated by the fixed behavior Q-function. In BQL, the behavior policy is modeled by MLP (Multi-Layer Perceptron). BDQL+BC represents additional behavior cloning constraint term has been added in Equation (8) to deal with the potential OOD issue. BDQL+SWA represents the BDQL along with the Stochastic Weight Averaging (SWA) that aims to stabilize the training fluctuation. BQL+SWA is the BQL with SWA.

**Diffusion Policy v.s. MLP Policy** Conclusion 4 claims that the optimality of BDQL lies in the small Total Varitional (TV) distance $D_{TV}\left(\pi_\theta \| \pi_b\right)$. We compare this distance implemented by diffusion with MLP via rliable package (Agarwal et al., 2021) in Figure 3 and find diffusion behavior policy significantly reduces this distance. In the Figure (2a,2b,2d,2e,2h,2i), BQL shows a noticeable decline in performance in the later stages of training. In contrast, BDQL stably reaches convergence, which also indicates the role of diffusion model as behavior policy.

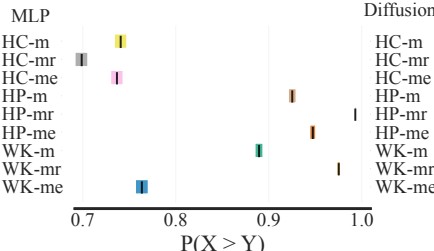

Figure 3: Total Varitional distance implemented by MLP is larger than Diffusion.

**SWA v.s. Policy Constraint** BDQL later training stage may suffer from intense training fluctuation or even collapse. For example, in Figure (2a,2b,2d), the BDQL is relatively stable while in Figure (2c,2e), BDQL training fluctuates and a sharp drop even occurs in Figure (2f). Policy constraint can make the training process (BDQL+BC) become very stable, such as Figure (2f), since the OOD issue has been completely overcame. However, the cost of stable training is that the constraint interfered with the original optimization direction, resulting in overall poor performance. In `Hopper` environment, Figure (2d,2e,2f), BDQL+BC hardly improves during the training. So we choose Stochastic Weight Averaging (SWA), which makes training robust to the rapid changes. Also in Figure (2d,2e,2f), BDQL+SWA not only stabilizes training but also improves gradually.

## 6.3 SPECIAL TRAINING PHENOMENON

We analyze the unique two-stage training process of BDQL. When $D_{TV}\left(\pi_{\tilde{\theta}} \| \pi_b\right) > 0.090$, BDQL suffers from OOD issue and collapse. When $D_{TV}\left(\pi_{\tilde{\theta}} \| \pi_b\right) < 0.090$, it has reached its peak performance, indicating offline RL is resolved before OOD occurred. For BDQL+SWA and BDQL+BC, the $D_{TV}$ is always less than 0.090 and the training is stable. This result well matches with theoretical analysis in section 4.

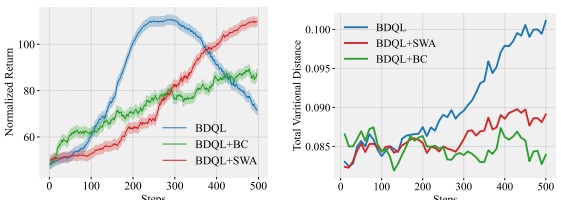

Figure 4: The special two-stage training phenomenon of BDQL (left) and distance $D_{TV}\left(\pi_{\tilde{\theta}} \| \pi_b\right)$ (right).

## 7 DISCUSSION, LIMITATIONS AND FUTURE WORK

For previous most offline algorithms, solving offline RL is inextricably linked with addressing OOD issue. But our proposed on-policy style offline algorithm Behavior Diffusion Q-Learning (BDQL) exhibits a novel two-stage training phenomenon where offline RL is solved before the OOD issue occurs. In other words, BDQL uniquely decouples the resolution of offline RL and the occurrence of OOD into two distinct training stages, which eliminates the necessity of policy constraints.

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

## A  PROOF OF THEOREM 1

*Proof.* We first prove $\mathbb{V}_{\pi_{\mathrm{b}}}(\mathbf{s}), \mathbb{V}_{\pi_{\mathrm{b}'}}(\mathbf{s}), \mathbb{V}_{\pi_{\mathrm{b}''}}(\mathbf{s}), \cdots, \mathbb{V}_{\pi_{\mathrm{b}*}}(\mathbf{s}) \leq \max\limits_{\substack{\mathbf{a} \in \mathcal{A} \\ \text{s.t. } \pi_{\mathrm{b}}(\mathbf{a}|\mathbf{s}) > 0}} Q^*(\mathbf{s}, \mathbf{a})$:

$$
\begin{aligned}
\mathbb{V}_{\pi}(\mathbf{s}) &= \mathbb{E}_{\substack{\mathbf{a} \sim \pi \\ \text{s.t. } \pi_{\mathrm{b}}(\mathbf{a}|\mathbf{s}) > 0}} Q_{\pi}(\mathbf{s}, \mathbf{a}) \\
&\leq \mathbb{E}_{\substack{\mathbf{a} \sim \pi \\ \text{s.t. } \pi_{\mathrm{b}}(\mathbf{a}|\mathbf{s}) > 0}} Q^*(\mathbf{s}, \mathbf{a}) \\
&\leq \mathbb{E}_{\substack{\mathbf{a} \sim \pi \\ \text{s.t. } \pi_{\mathrm{b}}(\mathbf{a}|\mathbf{s}) > 0}} \max_{\mathbf{a}} Q^*(\mathbf{s}, \mathbf{a}) \\
&= \max_{\substack{\mathbf{a} \in \mathcal{A} \\ \text{s.t. } \pi_{\mathrm{b}}(\mathbf{a}|\mathbf{s}) > 0}} Q^*(\mathbf{s}, \mathbf{a})
\end{aligned}
\tag{15}
$$

Replacing $\pi$ in the above inequality with $\pi_{\mathrm{b}}, \pi_{\mathrm{b}'}, \pi_{\mathrm{b}''}, \cdots, \pi_{\mathrm{b}*}$ completes the proof.

Secondly, we prove the improvement relation $\mathbb{V}_{\pi_{\mathrm{b}}}(\mathbf{s}) \leq \mathbb{V}_{\pi_{\mathrm{b}'}}(\mathbf{s}), \mathbb{V}_{\pi_{\mathrm{b}'}}(\mathbf{s}) \leq \mathbb{V}_{\pi_{\mathrm{b}''}}(\mathbf{s}), \cdots$. Maximizing $J(\pi_{\mathrm{b}}) = \mathbb{E}_{\mathbf{s}_t \sim \mathcal{D}, \mathbf{a}_t \sim \pi_{\mathrm{b}}} [Q_{\mathrm{b}}(\mathbf{s}_t, \mathbf{a}_t)]$ makes $\mathbb{V}_{\pi_{\mathrm{b}}}(\mathbf{s}) \leq Q_{\mathrm{b}}(\mathbf{s}, \pi_{\mathrm{b}'}(\mathbf{s}))$ and then we have

$$
\begin{aligned}
\mathbb{V}_{\pi_{\mathrm{b}}}(\mathbf{s}) &\leq Q_{\pi_{\mathrm{b}}}(\mathbf{s}, \pi_{\mathrm{b}'}(\mathbf{s})) \\
&= \mathbb{E}_{\pi_{\mathrm{b}'}} [r_t + \gamma V^{\pi_{\mathrm{b}}}(S_{t+1}) \mid S_t = s] \\
&\leq \mathbb{E}_{\pi_{\mathrm{b}'}} [R_t + \gamma Q^{\pi_{\mathrm{b}}}(S_{t+1}, \pi'(S_{t+1})) \mid S_t = s] \\
&= \mathbb{E}_{\pi_{\mathrm{b}'}} [R_t + \gamma R_{t+1} + \gamma^2 V_{\pi_{\mathrm{b}'}}(S_{t+2}) \mid S_t = s] \\
&\leq \mathbb{E}_{\pi_{\mathrm{b}'}} [R_t + \gamma R_{t+1} + \gamma^2 R_{t+2} + \gamma^3 V_{\pi_{\mathrm{b}'}}(S_{t+3}) \mid S_t = s] \\
&\vdots \\
&\leq \mathbb{E}_{\pi_{\mathrm{b}'}} [R_t + \gamma R_{t+1} + \gamma^2 R_{t+2} + \gamma^3 R_{t+3} + \cdots \mid S_t = s] \\
&= V_{\pi_{\mathrm{b}'}}(s)
\end{aligned}
\tag{16}
$$

Now we have $\mathbb{V}_{\pi_{\mathrm{b}}}(\mathbf{s}) \leq \mathbb{V}_{\pi_{\mathrm{b}'}}(\mathbf{s}) \leq \mathbb{V}_{\pi_{\mathrm{b}''}}(\mathbf{s}) \leq \cdots \leq \mathbb{V}_{\pi_{\mathrm{b}*}}(\mathbf{s}) \leq \max\limits_{\substack{\mathbf{a} \in \mathcal{A} \\ \text{s.t. } \pi_{\mathrm{b}}(\mathbf{a}|\mathbf{s}) > 0}} Q^*(\mathbf{s}, \mathbf{a})$. So, as policy updates, $\mathbb{V}_{\pi_{\mathrm{b}*}}(\mathbf{s}) \longrightarrow \max\limits_{\substack{\mathbf{a} \in \mathcal{A} \\ \text{s.t. } \pi_{\mathrm{b}}(\mathbf{a}|\mathbf{s}) > 0}} Q^*(\mathbf{s}, \mathbf{a})$. $\qquad \square$

## B  PROOF OF THEOREM 2

*Proof.* The key to proof lies in the clever application of the Hölder's inequality:

$$
\begin{aligned}
\left| J(\pi_{\mathrm{b}}) - \widehat{J}(\pi_\theta) \right| &= |\mathbb{E}_{\mathbf{s}_t \sim \mathcal{D}, \mathbf{a}_t \sim \pi_{\mathrm{b}}} [Q_{\mathrm{b}}(\mathbf{s}_t, \mathbf{a}_t)] - \mathbb{E}_{\mathbf{s}_t \sim \mathcal{D}, \mathbf{a}_t \sim \pi_\theta} [Q_{\mathrm{b}}(\mathbf{s}_t, \mathbf{a}_t)]| \\
&\leq \mathbb{E}_{\mathbf{s}_t \sim \mathcal{D}} |\mathbb{E}_{\mathbf{a}_t \sim \pi_{\mathrm{b}}} [Q_{\mathrm{b}}(\mathbf{s}_t, \mathbf{a}_t)] - \mathbb{E}_{\mathbf{a}_t \sim \pi_\theta} [Q_{\mathrm{b}}(\mathbf{s}_t, \mathbf{a}_t)]| \\
\text{(Hölder's inequality)} &\leq \mathbb{E}_{\mathbf{s}_t \sim \mathcal{D}} \|\pi_{\mathrm{b}}(\mathbf{a}_t|\mathbf{s}_t) - \pi_\theta(\mathbf{a}_t|\mathbf{s}_t)\|_1 \|Q_{\mathrm{b}}(\mathbf{s}_t, \mathbf{a}_t)\|_\infty \\
&= 2\mathbb{E}_{\mathbf{s}_t \sim \mathcal{D}} D_{TV}(\pi_\theta \| \pi_{\mathrm{b}})[\mathbf{s}_t] \cdot \max_{\mathbf{a}_t \in \mathcal{A}} |Q_{\mathrm{b}}(\mathbf{s}_t, \mathbf{a}_t)| \\
&\leq 2 \cdot \max_{\mathbf{s}_t \sim \mathcal{D}, \mathbf{a}_t \in \mathcal{A}} Q_{\mathrm{b}}(\mathbf{s}_t, \mathbf{a}_t) \cdot \mathbb{E}_{\mathbf{s}_t \sim \mathcal{D}} D_{TV}(\pi_\theta \| \pi_{\mathrm{b}})[\mathbf{s}_t]
\end{aligned}
\tag{17}
$$

$\qquad \square$

## C  ALGORITHM

---
**Algorithm 1** **B**ehavior **D**iffusion **Q**-**L**earning (BDQL)

---
1: Estimate behavior policy $\pi_\theta$ by Equation 6;
2: Calculate behavior $Q$-function $Q_\phi$ by Equation 7;
3: Update the optimal policy $\pi_{\theta*}$ by Equation 8;
4: Stochastic Weight Averaging (SWA) by Equation 14.

---

## D    EXPERIMENTAL DETAILS

**Software**    We use the following software versions:

- Python 3.8
- Pytorch 2.0.1 (Paszke et al., 2019)
- Gym 0.23.1 (Brockman et al., 2016)
- MuJoCo 2.3.7 (Todorov et al., 2012)
- mujoco-py 2.1.2.14

All the D4RL datasets (Fu et al., 2020) use the v2 version.

Our BDQL framework consists of two procedures: pre-training and fine-tuning. In the pre-training phase, it is important to highlight that the actor and critic are trained separately. The actor is trained using behavior cloning, where it learns by imitating the behavior of an expert actor based on their trajectory data. This training process focuses on teaching the actor to mimic the expert's actions and exhibit similar behavior. Simultaneously, the critic is also trained using the SARSA method with the same dataset. The critic estimates the value function for each state-action pair, allowing it to evaluate the quality of the actor's actions and provide feedback.

Consequently, in the fine-tuning phase following pre-training, the actor is refined with the assistance of the critic to attain an enhanced policy. This separation of training allows the actor and critic to fulfill distinct roles in the BDQL framework. The actor focuses on learning the desired behavior, while the critic provides guidance through rewards and corrections to improve the actor's policy. By emphasizing the individual training of the actor and critic, the BDQL framework combines their complementary strengths, ultimately leading to improved performance and more effective decision-making.

Table 2: Algorithm Parameters

|  | Hyperparameters | Value |
|---|---|---|
| BDQL | Optimizer | Adam (Kingma & Ba, 2014) |
|  | Critic learning rate | 3e-4 |
|  | Actor learning rate | 3e-4 |
|  | Mini-batch size | 256 |
|  | Discount factor | 0.99 |
|  | BDQL learning rate | 1e-5/1e-7 |
|  | actor pretrain steps | 1e6 |
|  | critic pretrain steps | 1e6 |
|  | Layer Normalization | True |
| Architecture | Critic hidden dim | 256 |
|  | Critic hidden layers | 3 |
|  | Critic activation function | Mish |
|  | Actor hidden dim | 256 |
|  | Actor hidden layers | 3 |
|  | Actor activation function | Mish |
| SWA | start step | 100 |
|  | update rate | 1 |
|  | SWA learning rate | 1e-5/1e-7 |

**Hyperparameters**    Our implementation of Diffusion-BC (Wang et al., 2022) is based on the author-provided implementations from GitHub. In the implementation, the BDQL policy is an MLP-based conditional diffusion model. Both the BDQL policy and the critic share the same MLP architecture, which consists of three-layer MLPs with Mish activation function and 256 hidden units. We also employ gradient normalization parameters from Diffusion-QL, which can be found in the same GitHub repository as Diffusion-BC. During the fine-tuning phase, the learning rate plays a crucial role. A larger learning rate can lead to quick collapse, so we set it to 1e-5. However, in the `Hopper-medium-expert` and `Halfcheetah-medium-expert` environments,

we found that the learning rate was too large, resulting in instability. Therefore, we set the learning rate to 1e-7 in these environments. The learning rate of the SWA is set to the same value as the regular BDQL learning rate. The SWA starts at step 100, and the update frequency is set to 1 step to ensure stable fine-tuning.

We outline the hyperparameters used by vary environments of D4RL in Table 2.

## E    EXPLORATORY DATA ANALYSIS

In BDQL framework, since the actor and critic are trained separately, it is necessary to explore the data distribution for each of them. We conduct separate analyses of the distribution of state-action and the distribution of state-reward.

### E.1    STATE-ACTION DISTRIBUTION

To enable easy visualization, the state data is reduced to two dimensions, while the action data is reduced to one dimension. Additionally, the datasets are standardized on a per-environment basis, ensuring consistency in the comparison process, and normalize both types of data to a range of 0 to 1. This approach enables easier and more meaningful comparisons between different environments. For the learning of Behavior Cloning (BC), a complex distribution of state-action pairs can lead to poor performance (Table 1). This can be observed from the fact that BC struggles to converge on the replay version dataset. The BC training phase of `HalfCheetah` shows the lowest overall standard deviation, which aligns with the distribution of state-action pairs depicted in the scatter plots (Figure 5a,5b,5c). In some other plots (Figure 5d,5g,5i), The same pattern is observed. A clearer distribution of state-action pairs leads to a more stable BC policy. On the other hand, for replay version dataset with complex data distributions (Figure 5e,5h), both performance and stability are negatively impacted. Based on the final experimental results (Table 1), it is evident that neither BQL nor BQL+SWA effectively converge when dealing with complex data distributions from the replay version dataset. However, BDQL outperforms the other methods on the replay version dataset. This indicates that by harnessing the modeling capability of Diffusion, BDQL can effectively address this issue and achieve improved performance.

### E.2    STATE-REWARD DISTRIBUTION

For the distribution of state-reward, the same dimensional reduction and normalization process is applied. The state data is reduced to two dimensions, while the rewards are normalized to a range of 0 to 1. By examining the distribution of state-reward pairs directly, we can gain insights into the learning dynamics of the rewards. This is particularly useful since the dataset stores data in the form of transitions. According to the results of BDQL (Table 1), during the fine-tuning stage, the critic demonstrates significant improvements in the `Walker2d` and `Hopper` environments. This suggests that a rich amount of reward information has been learned in these environments. The distribution of state-reward (Figure 6d,6e,6f,6g,6h,6i) supports this observation. In contrast, the improvements in the `HalfCheetah` environment are limited. It can be observed that the reward distribution in the `HalfCheetah` environment shows a significant bias (Figure 6a,6b,6c), contrasting with the other two environments where the reward distribution appears to be more uniform (Figure 6d,6e,6f,6g,6h,6i). This biased reward distribution has hindered the learning of critic. As a result, compared to the other two environments, the improvement brought by the critic in this case is limited. This calls for the need to propose better methods for modeling rewards, such as employing diffusion methods for BC. Unfortunately, no such method is presented at this time, and it will be left for future exploration.

Combining these distribution plots, we observe that in offline RL, the learning of behavior and reward face different challenges. By employing a two-stage approach, where the first stage focuses on addressing behavior learning and Q learning separately, the second stage integrates both estimated behavior and Q value, we can leverage this information more effectively and efficiently to tackle the problems encountered during learning. This approach aligns naturally with the inherent nature of the problem. In the future, further exploration of reward modeling and learning holds the potential to enhance the performance of our framework and make it more universally applicable. By delv-

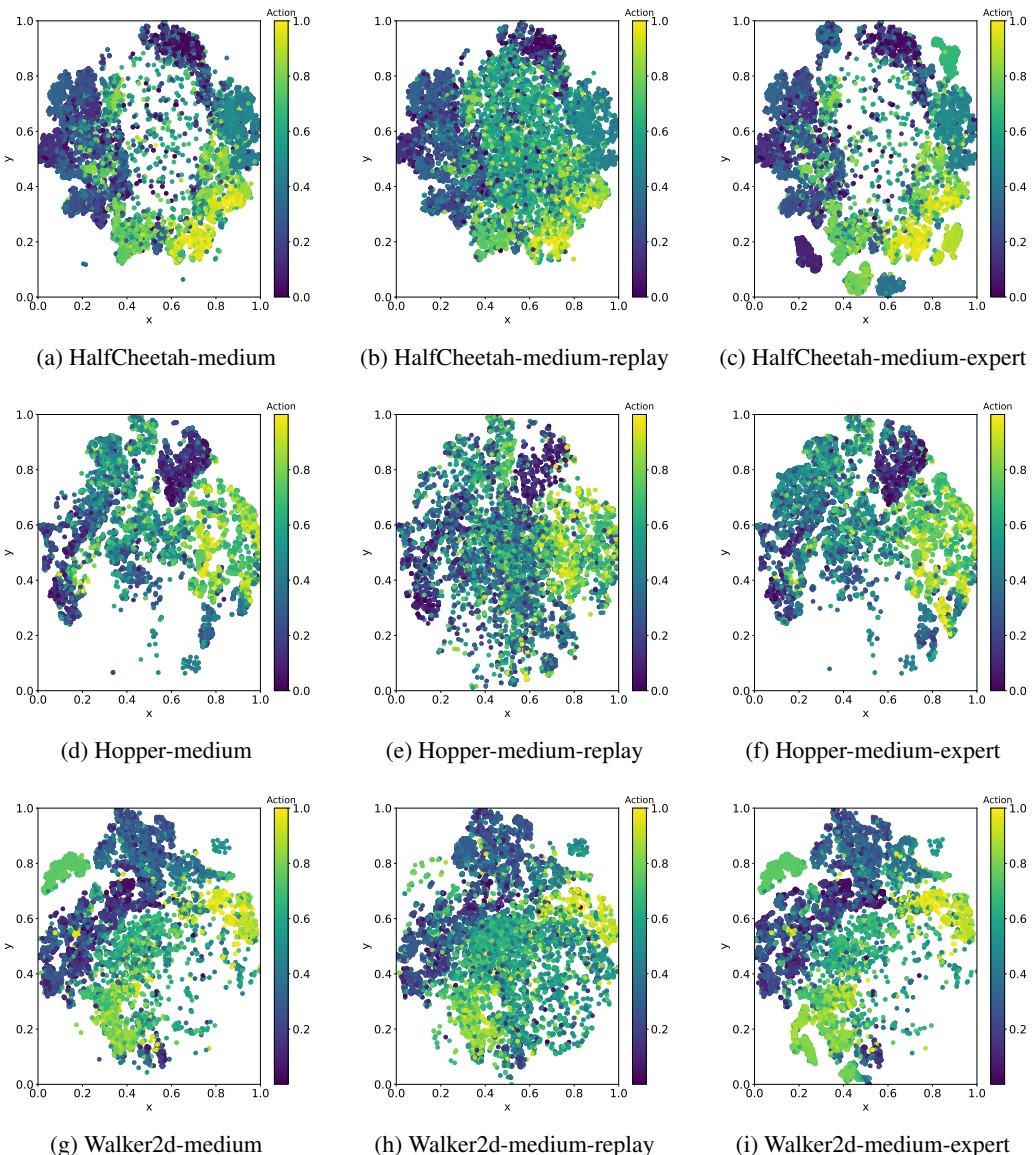

(a) HalfCheetah-medium    (b) HalfCheetah-medium-replay    (c) HalfCheetah-medium-expert

(d) Hopper-medium    (e) Hopper-medium-replay    (f) Hopper-medium-expert

(g) Walker2d-medium    (h) Walker2d-medium-replay    (i) Walker2d-medium-expert

Figure 5: The scatter plots display the distribution of state-action in the D4RL dataset. The state data is reduced to two dimensions, and the action data is reduced to one dimension. The x and y axes represent the two reduced dimensions of state data, while the color of the points represents the chosen action.

ing deeper into reward modeling and learning, we can potentially achieve better performance and broader applicability of our framework.

### E.3 TOTAL VARIATIONAL DISTANCE BETWEEN ESTIMATED BEHAVIOR POLICY AND OFFLINE DATASET

**Conclusion 4** indicates the optimality relates to the Total Variational distance between the estimated behavior policy and the true behavior policy. We empirically calculate this distance by replace the true behavior policy with offline dataset and then draw histograms in Figure 7. In comparison, the distribution of the diffusion is more concentrated and has a narrower range, while the MLP has a broader range. This also indicates that the diffusion more accurately models the behavior policy.

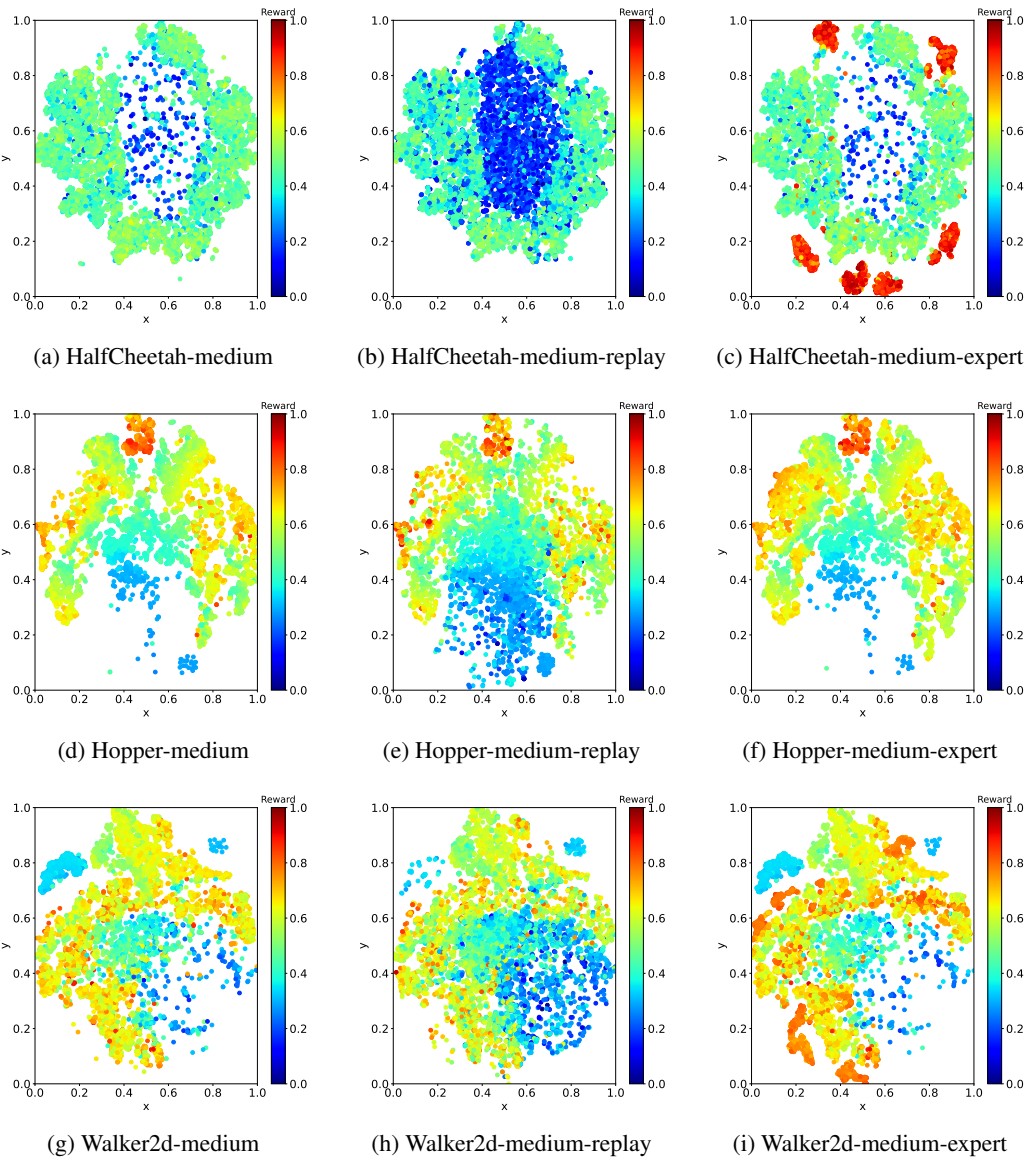

(a) HalfCheetah-medium

(b) HalfCheetah-medium-replay

(c) HalfCheetah-medium-expert

(d) Hopper-medium

(e) Hopper-medium-replay

(f) Hopper-medium-expert

(g) Walker2d-medium

(h) Walker2d-medium-replay

(i) Walker2d-medium-expert

Figure 6: The scatter plots display the distribution of state-reward in the D4RL dataset. The state data is reduced to two dimensions, and the color of the scatter plot represents the reward value. The rewards are normalized to a range of 0 to 1, where each reward corresponds to a specific state-action pair in the dataset.

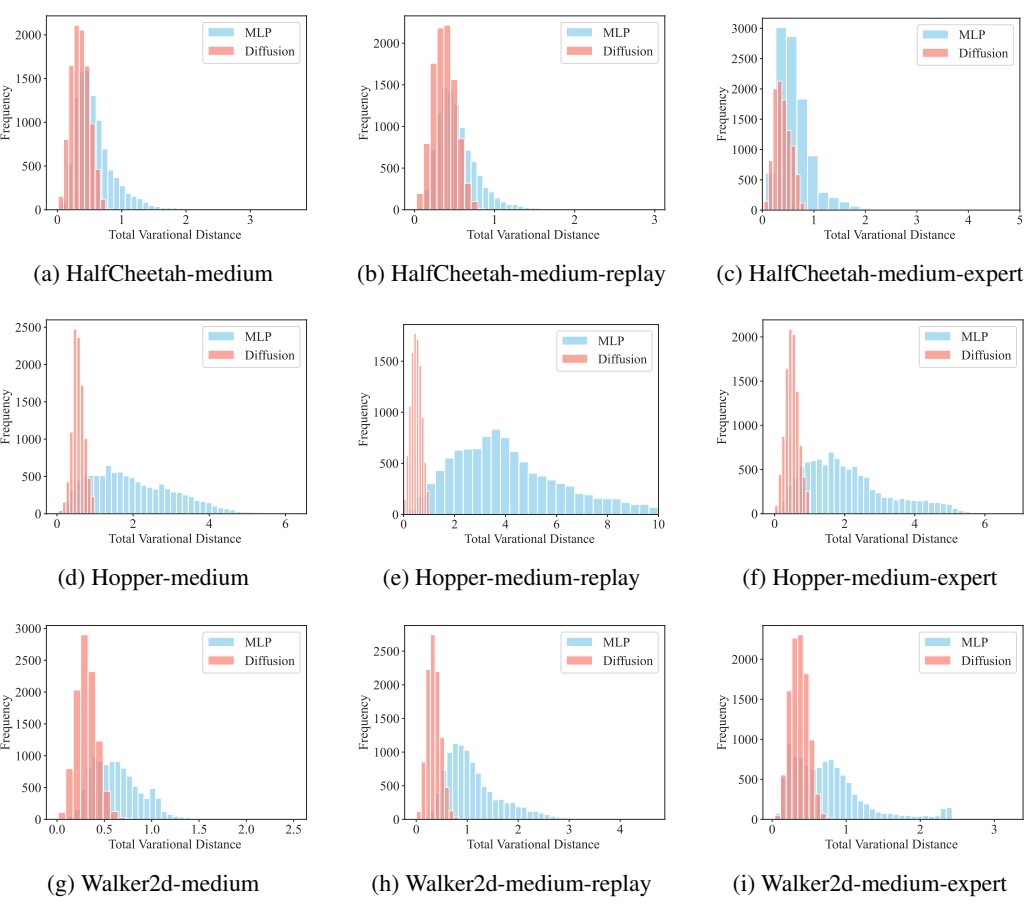

(a) HalfCheetah-medium

(b) HalfCheetah-medium-replay

(c) HalfCheetah-medium-expert

(d) Hopper-medium

(e) Hopper-medium-replay

(f) Hopper-medium-expert

(g) Walker2d-medium

(h) Walker2d-medium-replay

(i) Walker2d-medium-expert

Figure 7: The distribution of Total Varational distance between the estimated behavior policy (implemented by MLP or Diffusion) and offline dataset.

