# OpenReview forum: "BDQL: Offline RL via Behavior Diffusion Q-learning without Policy Constraint"
_ICLR.cc/2024/Conference — Submitted to ICLR 2024_

### Official Review · Reviewer_FiPY · 2023-10-20

**Soundness:** 3 good
**Presentation:** 3 good
**Contribution:** 2 fair
**Rating:** 5
**Confidence:** 4

**Summary:**

This paper discovers a good property of one-step RL (policy improvement with fixed behavior value function) equipped with an estimated diffusion-based behavior policy: due to accurate modeling of the behavior policy, one-step RL without any policy constraint can reach a strong enough performance before it suffers from OOD actions. To mitigate training fluctuation or collapse and stabilize the evaluation of learned policies, this paper introduces stochastic weight averaging of policy checkpoints. Experiments on MuJoCo tasks from D4RL demonstrate that the proposed BDQL-SWA provides good performance without policy constraints.

**Strengths:**

1. A good property of one-step RL with estimated diffusion behavior policies is discovered, which leads to a simple and clear algorithmic design.
2. Detailed ablation studies illustrate the contribution of each component.

**Weaknesses:**

1. The good property of two-stage training in BDQL is only validated in MuJoCo tasks, which are relatively simple. There is a lack of experiments on more complex datasets, such as AntMaze and Adroit domains from MuJoCo, or even heteroskedastic datasets [2]. Also, it is only validated empirically, without theoretical analysis.
2. The performance of BDQL-SWA on MuJoCo tasks still trails behind modern offline RL methods.
3. It is not the first time diffusion models have been utilized to model complex behavior distributions, which has been done by Chen et al. [1]. The authors should explicitly clarify this and sufficiently discuss the difference with them.

[1] Offline reinforcement learning via high-fidelity generative behavior modeling

[2] Offline RL With Realistic Datasets: Heteroskedasticity and Support Constraints

**Questions:**

1. Why 'The output of the diffusion policy is the deterministic action rather than the distribution of action'? If I understand correctly, the diffusion policy models a stochastic state-conditioned action distribution as Equation (3) rather than a single deterministic action. So Equation (8) should be $\mathbb{E}_{s_t \sim \mathcal{D}, a \sim \pi_\theta(s_t)} [Q_\phi(s_t, a)]$? Does this one still follow the deterministic policy gradient (DPG) theorem?
2. The name of 'theoretical performance' is inappropriate. Better names can be 'online validation performance,' 'best performance,' or 'ideal performance.'

---

> ### Author Response · Authors · 2023-11-21
> **Response to Reviewer FiPY**
>
> We express our gratitude for your comprehensive feedback. We've made careful considerations and addressed each of your concerns.
>
> **Q1: The good property of two-stage training in BDQL is only validated in MuJoCo tasks, which are relatively simple. There is a lack of experiments on more complex datasets, such as AntMaze and Adroit domains from MuJoCo, or even heteroskedastic datasets [2]. The performance of BDQL-SWA on MuJoCo tasks still trails behind modern offline RL methods.**
>
> **A1**: Although the results of BDQL are not outstanding, the experiments is achieved under the condition of "**without constraint to overcome OOD**". Combined with our newly added theoretical analysis, we comprehensively proves that BDQL is an algorithm capable of solving offline RL without the need for constraints.
>
> **Q2**: **Also, it is only validated empirically, without theoretical analysis.**
>
> **A2**: We have added some theory to better analyze the two-stage property of BDQL. In **Conclusion 1** of the new version of the paper, we find that the optimality of BDQL is based on the small enough distance (Total Varational distance between the estimated behavioral policy and the true behavior policy). Building on this conclusion, we divided the training stage based on the distance between the current training policy and the true behavior policy. We observed that the experiment perfectly matched this analysis.
>
> **Q3**: **It is not the first time diffusion models have been utilized to model complex behavior distributions, which has been done by Chen et al. [1]. The authors should explicitly clarify this and sufficiently discuss the difference with them.**
>
> **A3**: There is no apparent peculiarity in the way we use diffusion model. The crucial core is the unique role of diffusion policy. In the new version of the paper, we demonstrate through theoretical analysis (refer to **Conclusion 1** and the following description) that the theoretical framework of BDQL highly depends on the precise behavioral modeling provided by diffusion model.
>
> **Q4**: **Why 'The output of the diffusion policy is the deterministic action rather than the distribution of action'? If I understand correctly, the diffusion policy models a stochastic state-conditioned action distribution as Equation (3) rather than a single deterministic action. So Equation (8) should be $\mathbb{E}*{s_t \sim \mathcal{D}, a \sim \pi*\theta(s_t)} [Q_\phi(s_t, a)]$? Does this one still follow the deterministic policy gradient (DPG) theorem?**
>
> **A4**: From the details of the diffusion model, this action is sampled from a distribution. However, from an external perspective, the diffusion model takes in a state and outputs an action. This aligns with the definition of a deterministic policy and can be updated using DPG. Similar approaches are also adopted by Diffusion Q-learning [1].
>
> [1] Wang, Z., Hunt, J. J., & Zhou, M. (2022). Diffusion policies as an expressive policy class for offline reinforcement learning. arXiv preprint arXiv:2208.06193.
>
> **Q5**: **The name of 'theoretical performance' is inappropriate. Better names can be 'online validation performance,' 'best performance,' or 'ideal performance.'**
>
> **A5**: We have used "best result" in our new paper.
>
> Should there be any misinterpretation on our part regarding your inquiries or if we have not sufficiently resolved your concerns, please bring it to our attention at your earliest convenience. We look forward to any additional advice or guidance you can provide.

---

> > ### Comment · Reviewer_FiPY · 2023-11-22
> >
> > Thanks for the detailed response and additional theoretical analysis!
> >
> > However, I am not quite convinced by the theoretical analysis due to some unclear details.
> >
> > Q1: In (9) and (10), is $Q_{\pi_b}$ a ground-truth behavior Q function or just an estimated one? If it is the former, it is not the case in practical offline RL, since the training instability indeed comes from OOD action queries on estimated Q function. If it is the latter, why is there not any constraint in this offline objective (10) to avoid overfitting this estimated Q by learning OOD actions?
> >
> > Q2: When optimizing the objective (10), $\pi_b$ and $Q_{\pi_b}$ are both updated, right? But when optimizing the objective (9), $Q_{\pi_b}$ remains pre-defined, right? Thus,  $Q_{\pi_b}$ in the two are not the same one, do the derivation of Theorem 2 consider this? I highly recommend a more rigorous formulation of Section 4, elaborating on the detailed parameterization of each function (policy and Q),  and the parameters being optimized by each objective.
> >
> > Q3: This question is related to my weakness 1. There is no theoretical guarantee that a diffusion policy can approximate closely enough to the behavior policy, i.e., making $D_{T V}\left(\pi_\theta \| \pi_{\mathrm{b}}\right)$ small enough. In my opinion, complex datasets, listed in my original review, can pose a significant challenge to this. Thus, **my major concern** is that, the most important contribution of this paper, namely it is possible to do policy improvement without any policy constraint, may not generalize to more difficult offline RL scenarios beyond D4RL-Mujoco.
> >
> > Please correct me if there is any misunderstanding. Looking forward to the authors' response.

---

> > > ### Author Response · Authors · 2023-11-23
> > >
> > > We greatly appreciate the thorough review and valuable suggestions from the reviewers.
> > >
> > > >Q1: In (9) and (10), is $Q_{\pi_\mathrm{b}}$ a ground-truth behavior Q function or just an estimated one? If it is the former, it is not the case in practical offline RL, since the training instability indeed comes from OOD action queries on estimated Q function. If it is the latter, why is there not any constraint in this offline objective (10) to avoid overfitting this estimated Q by learning OOD actions?
> > >
> > > **A1**:In the theoretical analysis, $Q_{\pi_\mathrm{b}}$ represents the true behavior Q-function. This behavior Q is estimated through SARSA, which does not access out-of-distribution (OOD) actions. Therefore, no additional regularization is needed for the learning of this Q. In the subsequent learning process, this behavior Q is fixed and not updated.
> > >
> > > >Q2: When optimizing the objective (10), $\pi_\mathrm{b}$ and $Q_{\pi_\mathrm{b}}$ are both updated, right? But when optimizing the objective (9), $Q_{\pi_\mathrm{b}}$ remains pre-defined, right? Thus, $Q_{\pi_\mathrm{b}}$ in the two are not the same one, do the derivation of Theorem 2 consider this? I highly recommend a more rigorous formulation of Section 4, elaborating on the detailed parameterization of each function (policy and Q), and the parameters being optimized by each objective.
> > >
> > > **A2**: We sincerely apologize for lack of precision in our previous statements that may have led to confusion. In equation (10),$Q_{\pi_\mathrm{b}}$ is fixed which is the same as (9). We have provided a more rigorous proof in the updated version.
> > >
> > > >Q3: This question is related to my weakness 1. There is no theoretical guarantee that a diffusion policy can approximate closely enough to the behavior policy, i.e., making $D_{TV}\left(\pi_\theta|\pi_{\mathrm{b}}\right)$ small enough. In my opinion, complex datasets, listed in my original review, can pose a significant challenge to this. Thus, **my major concern** is that, the most important contribution of this paper, namely it is possible to do policy improvement without any policy constraint, may not generalize to more difficult offline RL scenarios beyond D4RL-Mujoco.
> > >
> > > **A3**: The insightful and valuable comment has given us an awareness of the need for more extensive experiments to validate the applicability of BDQL. During the rebuttal period, we attempted to include additional experiments. However, due to the slow inference speed of diffusion model, we are unable to obtain new experimental results. In the future, we plan to conduct more experiments for validation and explore optimizations to enhance the inference speed of diffusion policy.
> > >
> > > Once again, we appreciate the feedback from the reviewer, which provides clear directions for our next steps in improving the paper.

---

> > > > ### Comment · Reviewer_FiPY · 2023-11-23
> > > >
> > > > Thank you again for the response.
> > > >
> > > > Since I still cannot accurately grasp the theoretical analysis (for example, if $Q_b$ is fixed, what is optimized for objective (10), and what is the difference between (9) and (10)?), and there is no clear evidence addressing **my major concern** above,  I decide to keep my score.

---

> > > > > ### Author Response · Authors · 2023-11-23
> > > > >
> > > > > Both equation (9) and (10) updates the policy using the fixed behavior Q-function via deterministic policy gradient (DPG). The difference is the action distribution. The action in (9) is taken by the **estimated** bahvior policy (also the learned policy) while the action in (10) is taken by the **true** behavior policy.
> > > > >
> > > > > We appreciate your time, effort, and valuable suggestions. Thank you again.

---

### Official Review · Reviewer_BEcH · 2023-10-28

**Soundness:** 2 fair
**Presentation:** 3 good
**Contribution:** 2 fair
**Rating:** 3
**Confidence:** 4

**Summary:**

This manuscript studies the offline RL problem without policy constraint by utilizing the diffusion model as the tool. The authors further suggests the Stochastic Weight Average (SWA) to mitigate the training fluctuation. The superiority of the method is validated from D4RL tasks.

**Strengths:**

The introduction of diffusion model is a good trial for the offline RL community.  Although some papers have combined diffusion model in offline RL, this paper also devotes some ideas in this area.
The presentation is clear, and the paper is easy to follow.

**Weaknesses:**

1. First, from the experimental studies, the results of BDQL is not convincing. The performance of BDQL is close to other competitors and BDQL-SWA is even inferior to other competitors.
2. From the authors' explanation, the Theoretical performance refers to the on-policy scenario, and the Practical performance refers to the offline scenarios. In offline scenario, the performance of BDQL-SWA is still not good enough.
3. In SWA, the author mentions ``SWA averages the multiple checkpoints during the optimization''. Do you mean averaging the parameters of the models at different training time? So it is similar to the Target Actor/Critic network trick in most offline RL methods?

**Questions:**

1. Some offline-RL baselines, such as BEAR are missed in experiment, and the recent popular SPOT [1] method is not considered in experiments as well. It is suggested to consider the baselines in offline-RL methods.
2. The author claims that the BDQL has sufficiently solves the offline RL problem (OOD issue), and the OOD issue only causes fluctuation in training. The authors tries to illustrate this point with some ablation studies. However, this claim seems weak. It is suggested to add some theoretical analysis supporting this claim.
3. Actually, the diffusion model is time-costly in model inference. Will this issue also occur in BDQL? Some ablation studies on computation issues are suggested.
4. The review will consider increase the rating when some concerns are replied and solved.

[1] Supported Policy Optimization for Offline Reinforcement Learning. https://arxiv.org/abs/2202.06239

---

> ### Author Response · Authors · 2023-11-21
> **Response to Reviewer BEcH**
>
> Your detailed comments are greatly appreciated. We have carefully addressed each point, and your insights have contributed significantly to improving our work.
>
> **Q1**: **First, from the experimental studies, the results of BDQL is not convincing. The performance of BDQL is close to other competitors and BDQL-SWA is even inferior to other competitors**.
>
> **Q2**: **From the authors' explanation, the Theoretical performance refers to the on-policy scenario, and the Practical performance refers to the offline scenarios. In offline scenario, the performance of BDQL-SWA is still not good enough.**
>
> **A1&A2**: We address these two questions together. BDQL and its variants achieve results comparable to the strongest baseline (ensemble-based methods). While this may not be a  breakthrough,  it is important to note that this result is achieved **without introducing constraints and with OOD preserved**. This experiments, along with our newly added theoretical insights, demonstrate that offline RL may not necessarily require constraints to solve.
>
> **Q3**: **In SWA, the author mentions ``SWA averages the multiple checkpoints during the optimization''. Do you mean averaging the parameters of the models at different training time? So it is similar to the Target Actor/Critic network trick in most offline RL methods?**
>
> **A3**: Indeed, SWA averages different checkpoints during training, but it differs from the Target Actor/Critic network trick. The key distinction is that the Target Actor/Critic network actively participates in training, whereas SWA does not. SWA averages different checkpoints during training to stabilize fluctuations and avoid collapse.
>
> **Q4**: **Some offline-RL baselines, such as BEAR are missed in experiment, and the recent popular SPOT [1] method is not considered in experiments as well. It is suggested to consider the baselines in offline-RL methods**
>
> **A4**: The descriptions of the support constraint value function in BEAR and SPOT inspired our proof, and we have cited both of these papers. The instability of BEAR's performance is a major reason why we did not consider it in our baseline. The strength of the SPOT algorithm lies in its offline-to-online fine-tuning while the performance is comparable to TD3+BC in addressing offline RL (SPOT: 773/9 = 85.89, TD3+BC: 84.83). Meanwhile, our baseline also includes ensemble-based methods, which are among the strongest algorithms.
>
> **Q5**: **The author claims that the BDQL has sufficiently solves the offline RL problem (OOD issue), and the OOD issue only causes fluctuation in training. The authors tries to illustrate this point with some ablation studies. However, this claim seems weak. It is suggested to add some theoretical analysis supporting this claim.**
>
> **A5**: In the new version of the paper, we analyze the optimality of BDQL and its tenable conditions (please see Section 4). We find that when the estimated behavioral policy is close enough to the true behavioral policy, BDQL can converge to the optimal policy. Inspired by this conclusion, we divide the training process into two stages based on the distance between the current training policy and the true behavior policy. This leads to improved analytical results and better demonstrationl in our experiments (Figure 4).
>
> **Q6**: **Actually, the diffusion model is time-costly in model inference. Will this issue also occur in BDQL? Some ablation studies on computation issues are suggested.**
>
> **A6**: we compare the training time between BDQL and BQL. The training time of BDQL (1492 min) is approximately 2.5 times that of BQL (608 min). In future work, we will consider further optimizations in terms of algorithm and implementation to reduce the time overhead.
>
> In case our understanding of your questions is not accurate or if your concerns remain unaddressed, we encourage you to inform us promptly. We are keenly awaiting any additional suggestions or guidance you might offer.

---

> ### Author Response · Authors · 2023-11-23
>
> We would like to know if our response has addressed your concerns. Additionally, we are eager to hear your feedback on the newly added proof section. We look forward to your response and any further suggestions.

---

### Official Review · Reviewer_Chng · 2023-10-31

**Soundness:** 2 fair
**Presentation:** 3 good
**Contribution:** 1 poor
**Rating:** 3
**Confidence:** 3

**Summary:**

The current paper introduces a new offline RL algorithm using diffusion models for policy, named BDQL. The algorithm has three components: 1. it performs behavior cloning on the offline dataset, 2. critic learning by SARSA on the offline dataset, 3. policy improvement by deterministic policy gradient with the critic trained by SARSA, and stabilizing training with stochastic weight averaging. In the experiment, the paper compares BDQL with several baselines on the d4rl benchmark.

**Strengths:**

1. The paper lists important preliminaries so even readers who are not familiar with diffusion models can understand the context.

2. The ablation study is throughout.

3. The experiment compares with a variety of baselines.

**Weaknesses:**

1. The significance of section 2.2 is unclear, since both methods are not used in the current paper.

2. The SARSA update requires the assumption that the offline data is coming from trajectories, and the data collecting policy is a single stationary policy.

3. The choice of using SARSA to train the critic is actually confusing. According to the proposed algorithm, in the statistically asymptotical case, optimization is done perfectly, and offline data has good coverage, the critic will converge to the Q-function of the behavior policy, which might not be a strong policy, and the diffusion policy is just the argmax policy according to the Q-function of the behavior policy, which might be better than the behavior policy, but the performance is still not guaranteed. So it is hard to see why even in the most ideal setting this algorithm would return a strong policy.

4. The argument on regularization for ood from the current algorithm is not very convincing. It seems like an alternative way of regulizing with behavior cloning with a diminishing regularization coefficient. Using this perspective, one can also suspect if this varying objective is causing the instability in the practical performance.

5. No concrete algorithm box is provided.

6. The usage of the term "theoretical performance" in the experiment section is confusing.

**Questions:**

See above.

---

> ### Author Response · Authors · 2023-11-21
> **Response to Reviewer Chng**
>
> Thank you sincerely for your thorough and insightful comments. We have taken each question seriously and provided detailed responses.
>
> **Q1**: **The significance of section 2.2 is unclear, since both methods are not used in the current paper.**
>
> **A1**: In Section 2.2, we introduce on-policy and off-policy learning. The former is highly relevant to our algorithm since BDQL can be viewed as a form of on-policy learning in the context of offline RL. When introducing off-policy learning, we also highlight the challenges in offline RL. This section serves as essential groundwork for the subsequent discussions.
>
> **Q2**: **The SARSA update requires the assumption that the offline data is coming from trajectories, and the data collecting policy is a single stationary policy.**
>
> **A2**: SARSA can also be updated from transitions, not only trajectories. Offline datasets can be assumed to be collected by the behavior policy, which allows update by SARSA. This assumption has been adopted by on-policy style offline algorithms like Onestep RL [1] and BPPO [2].
>
> [1] Brandfonbrener, D., Whitney, W., Ranganath, R., & Bruna, J. (2021). Offline rl without off-policy evaluation. Advances in neural information processing systems, 34, 4933-4946.
>
> [2] Zhuang, Z., Kun, L. E. I., Liu, J., Wang, D., & Guo, Y. (2022, September). Behavior Proximal Policy Optimization. In The Eleventh International Conference on Learning Representations.
>
> **Q3**: **The choice of using SARSA to train the critic is actually confusing. According to the proposed algorithm, in the statistically asymptotical case, optimization is done perfectly, and offline data has good coverage, the critic will converge to the Q-function of the behavior policy, which might not be a strong policy, and the diffusion policy is just the argmax policy according to the Q-function of the behavior policy, which might be better than the behavior policy, but the performance is still not guaranteed. So it is hard to see why even in the most ideal setting this algorithm would return a strong policy.**
>
> **A3**: One of our innovations is updating the policy using only the critic **without introducing constraints**. Whether the learned policy is optimal or not can be addressed by our added proofs. In essence, when the policy estimated by BC is sufficiently close to the true behavior policy, the optimality of the policy is guaranteed.
>
> **Q4**: **The argument on regularization for ood from the current algorithm is not very convincing. It seems like an alternative way of regularizing with behavior cloning with a diminishing regularization coefficient. Using this perspective, one can also suspect if this varying objective is causing the instability in the practical performance.**
>
> **A4**: We may not fully understand this question and are unsure if the reviewer is asking if SWA is an alternative OOD regularization method. If so, our response is as follows; SWA does not participate in training and does not alter the original optimization objectives. This fundamentally distinguishes the role of SWA from other constraints. The training of the original BDQL is unstable, and SWA is used to eliminate this instability
>
> **Q5**: **No concrete algorithm box is provided.**
>
> **A5**: We have added the algorithm box in appendix and the Figure 1 can also clearly express the algorithm logic.
>
> **Q6**: **The usage of the term "theoretical performance" in the experiment section is confusing.**
>
> **A6**: We have replaced the “theoretical performance” with “best result”, representing the best result obtained from online evaluation.
>
> If we have misunderstood your queries or failed to address your concerns adequately, please do let us know promptly. We eagerly anticipate any further suggestions or guidance you may provide.

---

> > ### Comment · Reviewer_Chng · 2023-11-22
> > **Response**
> >
> > > A3: One of our innovations is updating the policy using only the critic without introducing constraints. Whether the learned policy is optimal or not can be addressed by our added proofs. In essence, when the policy estimated by BC is sufficiently close to the true behavior policy, the optimality of the policy is guaranteed.
> >
> > The contribution of performing offline RL without introducing constraints **with additional assumptions** may not be strong. For example, one can simply run fitted q iteration (FQI) when the data coverage has good coverage.
> >
> > The added theory part again seems to only provide a guarantee between the learned policy and the **behavior policy** (which again could be just a random policy), so it still seems unclear why this bound shows the optimality of the learned policy.
> >
> > I will keep my score at this point.

---

> > > ### Author Response · Authors · 2023-11-23
> > >
> > > Thank you for the thorough review of our additional proofs and your feedback. Here, we provide further explanation of our proofs and clarify some misunderstandings.
> > >
> > > >The contribution of performing offline RL without introducing constraints **with additional assumptions** may not be strong. For example, one can simply run fitted q iteration (FQI) when the data coverage has good coverage.
> > >
> > > It's not an assumption but a condition. Unlike the assumption of data convergence in FQI, BDQL doesn't make assumptions about the data. BDQL only requires that the **learned** behavior policy approximates the true behavior policy, which is achievable through diffusion (in Figure 3).
> > >
> > > >The added theory part again seems to only provide a guarantee between the learned policy and the **behavior policy** (which again could be just a random policy), so it still seems unclear why this bound shows the optimality of the learned policy.
> > >
> > > Theorem 1 proves the optimality of the theoretical form. Theorem 2 illustrates the distance between the theoretical and BDQL practical form. The combination of these two provides the guarantee of optimality for BDQL. Specifically, when the distance (between the learned behavior policy and the true behavior policy) is sufficiently small, the optimality of BDQL is ensured. The achievement of this small distance is realized through diffusion and is not theoretically guaranteed.

---

### Author Response · Authors · 2023-11-21
**Modification about the new paper version**

We would like to express our sincere appreciation to the reviewers for their insightful suggestions. All the comments have significantly contributed to further refining the paper. In this summary, we highlight the major modifications to facilitate a quick review of the new version.

We have added the proof for the optimality of the algorithm and the conditions under which it holds (section 4). Building upon these proofs, we have conducted a more thorough analysis of the two-stage nature of BDQL (section 4 and Figure 4) and provides clearer insights into the role of Diffusion model (Figure 3).

The performance of BDQL may not be outstanding. However, considering this is achieved **without constraints and with preserving OOD issue**, the result is quite impressive. We hope the reviewers take into account that BDQL is an algorithm attempting to solve offline RL without introducing constraints to overcome the OOD problem.

We sincerely hope that the reviewers will take the time to review the additional content and provide valuable feedback. We look forward to your critiques and suggestions.

---

### Meta-Review · Area_Chair_UgBU · 2023-12-05

**Metareview:**

This paper uses diffusion model to recover the behavior policy, and updates it using the behavior Q-function learned by SARSA.  The one-step RL works with no policy constraint and the training fluctuation is mitigated by stochastic weight averaging of policy checkpoints.

The reviewers find that several parts of the theoretical contributions need to be further clarified.  Although the rebuttal helped, I agree that the paper needs another round of review and revision before being published.

**Justification For Why Not Higher Score:**

Several parts of the theoretical contributions need to be further clarified.  Although the rebuttal helped, it is still not fully convincing.

**Justification For Why Not Lower Score:**

N/A

---

### Decision · Program_Chairs · 2024-01-16

Reject